# Genome-Wide Analysis of the Lateral Organ Boundaries Domain (LBD) Gene Family in *Solanum tuberosum*

**DOI:** 10.3390/ijms20215360

**Published:** 2019-10-28

**Authors:** Hengzhi Liu, Minxuan Cao, Xiaoli Chen, Minghui Ye, Peng Zhao, Yunyou Nan, Wan Li, Chao Zhang, Lingshuang Kong, Nana Kong, Chenghui Yang, Yue Chen, Dongdong Wang, Qin Chen

**Affiliations:** 1State Key Laboratory of Crop Stress Biology for Arid Areas, College of Agronomy, Northwest A&F University, Yangling 712100, Shaanxi, China; xnlhz@nwafu.edu.cn (H.L.); 472028412@qq.com (M.C.); 913259312@qq.com (X.C.); yeminghui@nwafu.edu.cn (M.Y.); zhaopeng@nwafu.edu.cn (P.Z.); 1462039919@qq.com (Y.N.); liwan@nwafu.edu.cn (W.L.); zhangchao520@nwafu.edu.cn (C.Z.); 18331121801@163.com (L.K.); knnnwafu@163.com (N.K.); 2018060035@nwafu.edu.cn (C.Y.); 2College of Food Science and Engineering, Northwest A&F University, Yangling 712100, China

**Keywords:** lateral organ boundaries domain, potato, gene expression, drought stress

## Abstract

Lateral organ boundaries domain (*LBD*) proteins belong to a particular class of transcription factors of lateral organ boundary (*LOB*) specific domains that play essential roles in plant growth and development. However, a potato phylogenetic analysis of the *LBD* family has not been fully studied by scholars and researchers. In this research, bioinformatics methods and the growth of potatoes were used to identify 43 *StLBD* proteins. We separated them into seven subfamilies: Ia, Ib, Ic, Id, Ie, IIa and IIb. The number of amino acids encoded by the potato LBD family ranged from 94 to 327. The theoretical isoelectric point distribution ranged from 4.16 to 9.12 Kda, and they were distributed among 10 chromosomes. The results of qRT-PCR showed that the expression levels of *StLBD2-6* and *StLBD3-5* were up-regulated under drought stress in the stem. The expression levels of *StLBD1-5* and *StLBD2-6* were down-regulated in leaves. We hypothesized that *StLBD1-5* was down-regulated under drought stress, and that *StLBD2-6* and *StLBD3-5* up-regulation might help to maintain the normal metabolism of potato and enhance the potatoes’ resistance to drought.

## 1. Introduction

The lateral organ boundaries domain (*LBD*) gene refers to a gene family with a special domain of lateral organ boundaries (*LOB*), also known as the asymmetric leaves2-like (*ASL)* gene family. In 2002, Shuai et al. first discovered *LOB,* which is expressed in a band of cells at the adaxial base of all lateral organs formed from the shoot apical meristem and at the base of lateral roots. [1]. *LBD* expression products represent a class of plant-specific transcription factors containing the LOB domain [2]. The LBD family has three specific protein domains which can be classified into Class I and Class II, according to their specific protein sequences, most of which belong to Class I [3,4]. They contain a similar zinc finger domain CX2CX6CX3C motif [1], a glycine-alanine-serine (GAS-block) region that it is similar to the leucine zipper-like domain, and a protein dimerization LX6LX3LX6L spiral coiled structure. Class II contains only structural motifs similar to the zinc finger CX2CX6CX3C [4]. Since the discovery of 43 LBD proteins in *Arabidopsis,* many LBD proteins have been found in plants such as *Oryza sativa*, *Malus domestica* and *Eucalyptus grandis Hill* [5,6,7,8,9,10,11,12,13,14,15]. Recently, researchers crystallized the LOB domain of TdBRaLD from the wheat *LBD* gene. This structure mainly consists of a zinc finger, a GAS motif consisting of two α-helices, a highly conserved five-residue motif (Asp-Pro-Val-Tyr-Gly, referred to as the DPVYG motif), and an amphipathic α-helix with the feature of a leucine zipper like coiled-coil element. The complex component between the monomer and the monomer determines the precise spatial arrangement of the two zinc fingers which are made up of two zinc fingers and combined with the palindrome [16].

The LOB domain plays a key regulatory role in the growth and development of plant organs and tissues [2]. Studies have shown that the zinc finger domain CX2CX6CX3C plays an important role in binding of the LBD protein to DNA. The GAS-block structure and LX6LX3LX6L helical coil structure are involved in the interaction between the LBD protein and other proteins. For example, the interaction between LBD and bHLH proteins can help reduce the affinity of the former to DNA [3,5]. Studies have confirmed that LBD proteins play an essential role in the regulation of the growth and development of a variety of plants. The *Arabidopsis AtLOB* (*AtASL4*) gene is mainly expressed at the proximal end of lateral tissues. It can be combined with the SHOOT MERISTEMLESS (STM) protein and BREVIPEDICELLUS (BP) protein in order to regulate the development of young leaves [1]. The overexpression of *AtLBD41* (*AtASL38*) in *Celosia plumosus* can causes leaf wrinkle deformity and has the phenomenon of obvious initialization. Therefore, it is speculated that it may be involved in regulation of the paraxial and abaxial polarity of the leaf [17]. The expression product of *AtLBD6* (*AtAS2*) inhibits cell proliferation and near-distal axis symmetry in the paraxial region of the leaf, forming a flattened leaf and regulating flower development through synergy with AS1 and JAG [18,19]. The LBD10 protein plays a crucial role in the development of *Arabidopsis* pollen [20]. *AtLBD6* (*AtAS2*) is a homologous gene of *OsAS2* in *Arabidopsis,* whose expression product can be involved in the regulation of rice bud differentiation and the border expression of different rounds indicating that this gene may regulate flower development [21]. *OsIG1* is involved in the flower organ number and gametogenesis in rice [22]. The overexpression of *OsLBD37* and *OsLBD38* can delay the process and speed of rice heading and increase yields [23]. The MdLBD13 protein can inhibit anthocyanin synthesis and nitrogen uptake in apple [24]. Maize *ZmIG1* regulates female gamete development and leaf axial differentiation [25]. The overexpression of eucalyptus *EgLBD37* increases the internode length, making plants higher, and increases lignin lignification components. The overexpression of *EgLBD29* reduces the length of the python and shortens the circular growth of the plant. Additionally, the overexpression of *EgLBD22* can increase the fiber content of the phloem [14]. *VvLBD19* was up-regulated in grape treated with 10% PEG for 24 h, which was 26 times higher than that of the control [26]. Zebarth, B.J, et al. found that the gene *St.LBD* (Unigene Stu.5076 from www.cpgp.ca) homologous to *AtLBD37*, *AtLBD38,* and *AtLBD39* in potato may be specifically in response to nitrate rather than N deficiency [27].

With the continuous development of sequencing techniques, many plant species have undergone genome sequencing, which lays a solid foundation for the use of bioinformatics research and identification of gene functions. Potato breeding is easy. Potato is an indispensable food crop in the world, and is the fourth largest food crop in China. However, the drought in northwestern China is not conducive to the growth of potatoes. Therefore, it is necessary to study and transform abiotic stress genes to improve the yields and quality of potatoes. A potato genome sequence has been published (PGSC, Potato Genome Sequencing Consortium, 2011) [28]. Potato grows rapidly and it has established a perfect genetic transformation system. It is hopeful that potato will become an autopolyploid model plant. Since the publication of the genome data of potato (*Solanum tuberosum* L.), there has been no complete report on the potato LBD transcription factor family. In this study, we use bioinformatics methods to identify 43 *LBD* family members from potato genome-wide sequences. A comprehensive analysis of basic information, physicochemical properties, chromosomal locations, genetic structures, evolutionary relationships, conserved domain features, and expression patterns has been conducted in this study, which can lay an important foundation for further studying the function of potato *LBD* genes.

## 2. Results

### 2.1. Identification of Potato LBD Gene Family Members

In this study, 43 *LBD* gene family members were identified (Table 1). The molecular weight and isoelectric point of potato LBD proteins were analyzed by the online website ExPASy (http://web.expasy.org/protparam/). The StLBD6-6 was the maximal protein, containing 327 amino acid residues, and the molecular weight was 37.4 KDa. The minimum protein was StLBD11-1, which has just 94 amino acid residues, and its molecular mass was 10.5 KDa. Their isoelectric point range was from 4.16 (StLBD11-1) to 9.12 (StLBD1-3) KDa. qPCR analysis showed that among the seven candidate genes, *StLBD1-5*, *StLBD2-6*, and *StLBD3-5* display significant differences under drought stress. We found that the *LBD* family of potato has the conserved nucleotide sequence “*CTCC*”, named “*CTCC-box*”.

### 2.2. Chromosome Localization of the Potato LBD Gene Family 

Through an analysis of the chromosomal localization, 43 *LBD* genes were distributed in 10 of the 12 chromosomes (Figure 1). Among them, chromosome 6 (Chr06) had the most indispensable genes distributed. It contained 12 *LBD* genes, followed by chromosome Chr02, containing 9 *LBD* genes. The Chr05, Chr08, and Chr12 chromosomes contained only one *LBD* gene. The Ch07 and Ch10 chromosomes did not exhibit distribution of the *LBD* gene. The six pairs of genes are closely linked together with the chromosome (*StLBD1-2*/*StLBD1-3, StLBD2-1*/*StLBDD2-2*, *StLBD2-6*/*StLBD2-7*, *StLBD6-2*/*StLBD6-3*, *StLBD6-6*/*StLBD6-7*, and *StLBD9-3*/*StLBD9-4*) and belong to the potato paralogous gene pair.

### 2.3. Phylogenetic Evolution and Gene Structure Analysis of the Potato LBD Gene Family

For analysis, we constructed a phylogenetic tree of the potato LBD (Figure 2). The results showed that 43 potato *LBD* genes were divided into Class I and Class II, which contained 35 and 8 *LBD* genes, respectively. Twenty-six potato *LBD* genes formed 13 pairs of paralogs in which nine pairs of genes had a bootstrap higher than 90: they were *StLBD2-1*/*StLBD2-2*, *StLBD3-3*/*StLBD6-7*, *StLBD6- 12/StLBD9-3*, *StLBD11-2*/*StLBD11-3*, *StLBD6-1*/*StLBD6-10*, *StLBD6-4*/*StLBD6-9*, *StLBD11-1*/*StLBD12-1*, *StLBD9-1*/*StLBD9-2,* and *StLBD1-5*/*StLBD2-6*. For further analysis of the evolutionary relationship between the St*LBD* and *AtLBD*, we conducted specific research on a phylogenetic tree of the potato and *Arabidopsis* LBD system (Figure 2). The results of phylogenetic tree clustering show that Class I was subdivided into five subclasses of Ia, Ib, Ic, Id, and Ie, which respectively contained 8, 7, 16, 0, and 4 potato LBD family members. *Arabidopsis* contained 8, 13, 5, 2, and 8 LBD family members. Class II was subdivided into two subclasses: IIa and IIb. Potato contained five and three members in IIa and IIb, respectively. *Arabidopsis* contained four and three members in IIa and IIb, respectively. Interestingly, our results were roughly the same as those of the incomplete potato study conducted by Bdeir et al. [29]. The difference is that we found seven homologous genes of potato to *AtLBD1,* five of which had higher similarities: *StLBD4-1*, *StLBD6-8*, *StLBD6-2*, *StLBD11-2*, and *StLBD11-3*. Furthermore, two, including *StLBD6-11* and *StLBD1-1,* had low similarities. Additionally, *StLBD1-3,* a homologous gene of *AtLBD18*, has not been found by predecessors. Gene structure analysis (Figure 3) showed that the potato *LBD* genes have a simple structure and no more than two introns. Only two genes contained two introns, while 29 genes contained one intron and 12 genes had no introns. The results of cluster analysis showed that similar cluster relations have similar gene structures. For example, both StLBD3-4 and StLBD9-2 have conformable conserved domains. In addition, six genes in Class II are very similar. They only have one intron and identical motifs. 

### 2.4. Conservative Analysis of the Potato LBD Protein Sequence

By blasting the LOB domains of 43 potato LBD proteins (Figure 4), most of the protein sequences had a CX2CX6CX3C motif similar to the zinc finger structure at the N-terminus. Among them, five proteins (StLBD11-2, StLBD1-1, StLBD1-2, StLBD9-1, and StLBD6-11) did not contain the CX2CX6CX3C motif. There were four amino acids between the third and fourth cysteines of StLBD2-1. Four proteins (StLBD2-8, StLBD3-4, StLBD6-6, and StLBD8-1) lacked the third cysteine. A total of 10 LBD protein sequences (StLBD6-8, StLBD12-1, StLBD3-1, StLBD2-3, StLBD2-9, StLBD3-3, StLBD6-7, StLBD3-2, StLBD6-12, and StLBD9-3) contained complete proteins similar to the leucine zipper LX6LX3LX6L motif. Twenty-nine of the 43 potato LBD proteins contained a complete GAS-block between the two motifs CX2CX6CX3C and LX6LX3LX6L. With the exception of StLBD6-3, the remaining genes contained a conserved proline (P) on the right side (C-terminus) of this module. It is speculated that this module plays a significant role in performing biological functions. In *Solanum tuberosum*, glycin was highly conserved in the GAS module, as it was against *Populus euphratica*, *Daucus carota,* and *Capsicum annuum*. In addition to StLBD6-3, there was also highly conserved arginine (R) and glycine (G) at the C-terminus of all proteins. This suggested that these three amino acids might have some special biological functions in the growth and development of potato. 

### 2.5. Tissue Expression and Induced Expression Analysis of the Potato LBD Gene

To explore the function of the potato *LBD* gene during the process of growth and development, we obtained transcriptome data from the potato genome sequencing consortium (PGSC) database (http://solanaceae.plantbiology.msu.edu/) [28] for tissue expression and stress-induced expression of the potato *LBD* gene. Then, we mapped the potato *LBD* gene family in different tissues: the flower, leaf, petiole, root, shoot apex, stamen, stem, stolon, tuber cortex, tuber peel, tuber pith, tuber sprout, mature tuber, young tuber, and whole plants. Different treatments (*β*-aminobutyric acid, methyl benzophenazole thioacetate, Phytophthora infestans, sodium chloride, mannitol, heat stress, abscisic acid, gibberellin, 6-benzylaminopurine, and auxin) provided their own respective expression on the heatmap (Figure 5 and Figure 6). Blue and white indicate the intensity of the gene in the heatmap. A brighter blue color indicates a higher expression level, while a brighter white color indicates a lower gene expression level. The results showed that *StLBD1-2*, *StLBD1-5*, *StLBD2-1*, *StLBD2-2*, *StLBD2-3*, *StLBD2-6*, *StLBD2-7*, *StLBD2-9*, *StLBD3-1*, *StLBD3-2*, S*tLBD3-4*, *StLBD3-5*, *StLBD4-2*, and *StLBD6-5* were mainly expressed in the flower, and *StLBD3-5* had the highest expression in the flower. *StLBD1-5*, *StLBD2-4*, *StLBD2-6*, *StLBD3-5*, *StLBD6-4*, and *StLBD11-1* were highly expressed in the root. *StLBD3-5* and *StLBD11-1* had a higher expression in the root than other genes; they may be involved in the developmental regulation of the root. The expression of *StLBD1-5*, *StLBD2-6*, *StLBD3-2*, *StLBD3-5*, *StLBD4-2*, *StLBD6-4*, *StLBD11-2*, *StLBD11-1* and *StLBD12-2* was up-regulated in the stem relative to other genes. *StLBD2-6* and *StLBD3-2* had the highest expression in the stem, suggesting that these genes might be involved in stem growth regulation. *StLBD2-6*, *StLBD2-9*, *StLBD3-5* and *StLBD11-1* were highly expressed in potato tuber peel. The expression level of *StLBD3-5* was regarded as the highest, suggesting that this gene plays a certain role in the development of potato tuber peel. As is shown above, *StLBD3-5* had a certain relationship with potato tuber peel terms of color formation. The expression level of StLBD3-5 was also high, but there was no significant difference in mature tuber and young tuber. This indicated that this is an important regulatory gene for tuber growth. Other genes had different expression levels at different stages of potato tuber formation. For example, the expression level of *StLBD6-3* increased in the early stage of tuber formation while it was not expressed after tuber maturation, showing that *StLBD6-3* is involved in the development of tuber. The expression of *StLBD1-2* was down-regulated in the early stage of tuber formation, but was not expressed after tuber maturation, which may show that this gene works against tuber development. *StLBD2-9* exhibited a low expression level during the early stage of tuber formation, and which the expression increased during tuber maturation. This indicates that this gene is involved in the tuber maturation process. The expression of *StLBD3-2* turned out to be positively increased from the early stage of tuber formation but was negatively expressed in the late stage of tuber maturation. This shows that this gene is involved in the initial formation of tuber. *StLBD3-2* had nothing to do with tuber expansion. *StLBD2-3*, *StLBD2-4*, *StLBD2-5*, *StLBD2-6*, *StLBD2-9*, *StLBD3-2*, *StLBD3-5*, *StLBD11-1,* and *StLBD11-2* were highly expressed in the buds of tuber. The *StLBD3-5* was up-regulated in young tuber and mature tuber, displaying no significant difference between the two, which shows that this gene plays a key role in the growth of tuber. *StLBD2-6* and *StLBD3-5* had high expression levels in the whole plant, indicating that these genes play an important role in the growth and development of potato throughout the period. *StLBD1-1*, *StLBD1-4*, *StLBD2-8*, *StLBD4-1*, *StLBD6-11*, *StLBD8-1,* and *StLBD9-2* all expressed a low or down-regulated expression in various tissues and organs. Various stress treatments were tested for the doubled monoploid potato variety (DM), for which the expression levels of some genes significantly increased. The *StLBD11-1* was upregulated at 24, 48, and 72 h after induction with phenylthiophene thioacetate (BTH) and at 24, 48, and 72 h after induction with *P. infestans*. This showed that *StLBD11-1* responded to the plant immune mechanism. The expression level of *StLBD1-4* and *StLBD9-1* genes were significantly increased under stress of 150 mM sodium chloride for 24 h, which showed that these genes were involved in the salt tolerance response of potato. After 260 μM mannitol treatments for 24 h, the expression levels of *StLBD1-2* and *StLBD4-1* were increased, showing that these two genes respond to the drought resistance mechanism in potato. After treatment for 35 degrees heat stress for 24 h, the expression levels of *StLBD4-2*, *StLBD6-2*, *StLBD6-11*, *StLBD9-3,* and *StLBD9-4* genes were relatively high, indicating that these genes were related to the heat resistance of potato. Under 24 h stress treatment of 50 μM GA3, the expression levels of *StLBD1-4*, *StLBD6-11,* and *StLBD9-4* were increasing. The expression levels of *StLBD1-4*, *StLBD4-1*, *StLBD6-11,* and *StLBD9-4* were increased under the 24 h treatment of 10 μM indole-3-acetic acid (IAA), suggesting that these genes play a crucial role in the regulation of potato growth and development.

### 2.6. Analysis of Cis-Acting Elements of the Potato LBD Gene

To further understand the transcriptional regulation of the identified potato *LBD* family genes, we need to further understand the promoter region. We used the online prediction software Plant CARE (http://bioinformatics.psb.ugent.be/) to search the cis-acting elements of the potato *LBD* genes in the vicinity of the upstream 1000 bp region. In this study, 63 cis-acting elements were identified from 43 potato LBD genes (Figure 7). Except for *StLBD9-4*, which does not have CAAT-box, the other genes were shown to contain regulatory elements such as CAAT-box and TATA-box. Among them, CAAT-box is a cis-acting element commonly found in the promoter and enhancer regions of many eukaryotic genes [30]. It is a core promoter element 30 bp upstream of the TATA-box transcription initiation site which plays a crucial role in the precise localization of transcription initiation [31]. In these cis-acting elements, I-Box, Box4, G-Box, GT1-motif, GATA-motif, and TCT motifs can respond to different frequencies of light, controlling plant growth, development, the circadian rhythm, and nitrate uptake. GATA-motif is an important domain of the plant light-regulated response which has three GATA-motif repeats on the promoter of the *Petunia hybrida* chlorophyII-binding protein gene (Cab22) [32]. There are many light-regulated genes in the GT1-motif, such as rice and oat *PHYA,* which are ubiquitous in the GT-1 binding site [33]. G-box is involved in photoreaction regulation and its binding factors are usually members of the bHLH, bZIP, and NAC families [34]. In addition, there are many important cis-acting elements in the promoter region of the *StLBD* genes. For example, the repetitive structure of a large number of TC can respond to defense responses and stress. TGACG-motif and CGTCA-motif are involved in the methyl jasmonate response. Long terminal repeat (LTR) is related to heat shock reactions. Adenine and uridine-rich elements (ARE) and GC-motif are involved in anaerobic induction. O2-site is a glutinin metabolism regulating element. The myeloblastosis (MYB) binding sites (MBS) respond to abscisic acid (ABA) signaling and are involved in drought induction. Circadian is associated with circadian rhythm control [35]. When plants are affected by drought stress, the content of ABA will increase. ABA can promote stomatal closure, reduce transpiration, and alleviate water deficit in plants. It was shown through analysis that several potato *LBD* genes contain ABA responsive element (ABRE) cis-acting elements, namely ABA response elements [36]. These are an important component of the potato *LBD* gene response to ABA under adverse conditions. During the analysis, we found an unnamed cis-acting element “Unnamed_4”, whose nucleic acid sequence is “CTCC”. We named it the “CTCC-box”. It has been shown that “CTCC” can completely eliminate the expression of the *DYT1* gene, thereby preventing the formation of tapetum. The results indicate that the “CTCC” sequence is indispensable for the normal expression of *DYT1* [37]. There are currently no reports on its features. Except for the *StLBD3-3* all of the potato *LBD* contained this structure. This shows that this structure has a significant role in transcription initiation of the *LBD* gene family.

### 2.7. Expression Analysis of Seven LBD Genes in the Leaf at Different Stages

We focused on several potato *LBD* genes that have a high homology to the *Arabidopsis LBD* gene previously studied. They were *StLBD1-5*, *StLBD2-6*, *StLBD3-1*, *StLBD3-5*, *StLBD6-5*, *StLBD3-2*, and *StLBD11-2*. In order to prove the regularity of the seven observed potato *LBD* genes under an arid condition, qRT-PCR was used to detect the expression level of growing potato seedling leaves. The physiological state of the material, which was recorded six times, is shown in Figure 8. The qRT-PCR data was processed using the 2^−ΔCT^ method. The results of qRT-PCR showed that each gene had a different expression level on different days and under different treatment conditions. (Figure 9). Within 12 days after treatment, the expression level of *StLBD1-5* in the drought stress and watering group displayed no significant differences. On the 16th day, the potato began to show significant differences under an arid condition (Figure 8K,E). Responses of the different treatments of the *StLBD1-5* began to show significant differences after the 16th day, indicating that under an arid condition, the expression level of *StLBD1-5* is significantly inhibited (Figure 9A). Over time, the expression level of *StLBD2-6* was similar to *StLBD1-5*. After 16 days, these two treatments began to show significant differences and a down-regulated expression. Different from *StLBD1-5*, *StLBD2-6* had the highest expression on the 16th day, showing that the value was greater under a moist condition than under an arid condition (Figure 9B). The expression level of *StLBD3-1* was very low in the leaf. The expression level of *StLBD3-1* was the highest on the 12th day under an arid condition and the seedling was not lodged at this time (Figure 8J). Its expression was significantly higher than the watering group, but it was down-regulated after the 12th day (Figure 9C). This indicated that a certain degree of arid condition can also induce the expression of this gene. Similarly, *StLBD3-2* might also be induced to expression under an arid condition. On the 8th day, the expression level of *StLBD3-2* was significantly higher than the watering group and was then down-regulated (Figure 9D). This shows that the gene responds to drought stress, but excessive drought stress is unhelpful to the expression of this gene. As the plant grew, the expression level of StLBD3-5 was up-regulated (Figure 9E). On the 16th day, the expression level of *StLBD3-5* in the watering group was significantly higher than that in the drought stress group. A certain amount of drought stress may increase the expression level of *StLBD6-5*. On the 12th day, the expression level of *StLBD6-5* in the watering group was significantly higher than in the drought stress group and was then down-regulated (Figure 9F). This indicated that the expression of this gene was almost unaffected by drought stress and mainly changed periodically. It can be clearly seen that the expression of the *StLBD11-2* gene fluctuates upwards every 8 days. It seems to follow a certain periodic pattern, where the gene hardly responds to drought stress (Figure 9G). This suggests that this gene might be associated with certain physiological cycle changes in potato leaf.

### 2.8. Expression Analysis of Seven LBD Genes in Different Tissues

In order to prove the change in the *LBD* gene expression level of different tissues of potato under drought stress, qRT-PCR was used to analyze the expression level of potato treated for 20 days. The physiological state is shown in Figure 10. The results of qRT-PCR showed that the expression of *StLBD1-5* in the leaf of the watering group was four times higher than that in the drought stress group. The expression level of *StLBD2-6* is approximately more than two times higher in the watering group then in the drought stress group. This shows that under a normal watering condition, *StLBD1-5* and *StLBD2-6* are essential for maintaining the physiological metabolism of potato leaf. Drought stress significantly inhibited the expression of *StLBD1-5* and *StLBD2-6* genes in potato leaf (Figure 11A). It should be emphasized that the leaves wrinkled and wilted after 20 days of drought stress in the drought stress group, but retained a certain amount of water, exhibiting cell viability and tubers (Figure 10B). In tuber, the expression of *StLBD3-5* was significantly higher than the other six genes’ expression of *StLBD3-5* in the drought stress group than in the watering group, but there was no significant difference (Figure 11B). This period is in the stage of tuber initiation, so *StLBD3-5* may play a key role in potato tuberization. Among the stems of potato, the expression level of S*tLBD2-6* was the highest under an arid condition, which was three times that under the moist condition (Figure 11C). The expression of *StLBD3-5* under drought stress was second only to the expression of *StLBD2-6* under drought stress. The expression of *StLBD3-5* in the drought stress group was six times that of the watering group (Figure 11C). This indicated that the arid condition could significantly increase the expression level of *StLBD2-6* and *StLBD3-5* in the potato stem. These two genes might have a positive effect on maintaining physiological metabolism in the potato stem when the potato is subjected to drought stress. In the root of potato, the expression levels of each gene were very low. The expression level of *StLBD2-6* in the watering group was the highest, being higher than that in drought stress group, indicating that drought stress could adversely affect the expression of the *StLBD2-6* gene but it did not show a significant difference. The expression level of *StLBD1-5* was also much higher than that of the other five genes and the expression of this gene was hardly affected by drought stress (Figure 11D). It should be emphasized that the water content in the root of the drought stress group and watering group was very different. The root of the drought stress group was obviously dry. However, there was no difference in the expression level of the *StLBD1-5* gene between the two groups. It is worth mentioning that *StLBD3-1* was very weakly expressed in these four tissues of potato and can hardly be identified from Figure 11. From the expression data of *StLBD3-1*, the expression of the gene in the stem increased under an arid condition. 

## 3. Discussion

Potato is one of the most important non-cereal crops in the world. It feeds a large portion of the world. It is also China’s fourth largest food crop. It has many advantages, such as a high yield, delicious taste, and the combined use of food and grain. It is important to northern China’s cultivated crops. In 2011, the potato (*Solanum tuberosum* L.) genome sequence and related functional data were published (PGSC, 2011), which enabled the genome-wide identification of *StLBD* family genes and corresponding transcripts. To date, several gene families in potato have been identified and analyzed, such as *PHT* [38], *Aux/IAA* [39], *Hsp20* [40], *bHLH* [41], *WRKY* [42], etc. However, identification of the potato *LBD* gene family at the genomic level has not been fully reported. The LBD protein family is a plant-specific transcription factor that plays an important role in plant growth, development, and adversity stress [1,4,43,44]. In this study, 43 *StLBD* gene have been discovered in the potato genome (Table 1), which can be divided into two major classes and seven subclasses (Ia~Ie, IIa, and IIb). The number of family genes was 43 for *Arabidopsis* [3], 44 for *Zea mays* [9], 45 for *Capsicum annuum* [11], 57 for *Populus trichocarpa* [45], 45 for *Nicotiana tabacum* [46], 35 for *Oryza sativa* [47], etc. This indicated that the *LBD* family retained a largely fixed function of the genetic evolution of different species. Some functions have been verified in several LBD members of the model plant *Arabidopsis*. In this research, StLBD homologous proteins were obtained by a cluster analysis of the potato protein sequence and AtLBD protein sequence. Many homologous genes (Figure 2) provided a convenient and reliable theoretical basis for predicting the function of the *StLBD* family. It is noteworthy that the StLBD family lacks “Id” members and they might have been lost in the process of evolution. Analysis by chromosomal localization revealed that two chromosomes, Ch07 and Ch10 have not been found in the *LBD* gene. Additionally, three *LBD* genes were dispersed on one chromosome, which were *StLBD5-1* on Ch5, *StLBD8-1* on Ch8, and *StLBD12-1* on Ch12. The other *LBD* genes were dispersed on the other 10 chromosomes (Figure 1). The state of these gene distributions was similar to that of previously reported model plants of *Arabidopsis*. Chromosomal localization analysis of the potato *LBD* gene family revealed that there were six homologous gene pairs, which indicated that the gene family occurred by tandem gene duplications during evolution (Figure 1). The *StLBD* gene had a simple structure of no more than two introns and adjacent genes had similar intron numbers. MEME program predictive analysis indicated that *LBD* gene members with similar evolutionary relationships have similar structures (Figure 3). The tissue expression model and functional characteristics of genes were closely related. Our study used potato tissue expression transcript data to analyze the expression of the St*LBD* family in 15 tissues (Figure 5). We also detected the gene expression levels of seven potato *LBD* genes in the leaf, stem, tuber, and root that were highly homologous to the *Arabidopsis LBD* gene by qRT-PCR (Figure 11). Experimental results were basically similar to the reported transcript data. However, the expression levels of *StLBD3-2* and *StLBD11-2* were lower than the reported transcript data in the leaf in this study. We found that the expression levels of *LBD* exhibited a significant difference in different tissues. The expression of some genes showed significant differences under an arid condition. Anthocyanins play a key role in the plant defense [27]. In previous studies, *AtLBD37*, *AtLBD38,* and *AtLBD39* were negative regulators of anthocyanin synthesis. The overexpression of any of these three genes in the absence of N/NO_3_- strongly suppresses the key regulators of anthocyanin synthesis [48]. Therefore, we conclude that a high expression of these *LBD* was detrimental to increasing plant stress tolerance. The *DFR* gene has been shown to be an enzyme in the anthocyanin biosynthesis pathway [49]. With the supply of high nitrates, *DFR* expression was reduced. Anthocyanin accumulation was reduced by Zebarth, B.J. et al [27]. This was associated with an increased expression of the N-regulated anthocyanin synthesis inhibitor *St.LBD* (Unigene Stu. 5076 from www.cpgp.ca). However, ammonium increase has no effect on the expression of *St.LBD*, showing that the gene may have a specific effect on nitrate rather than N deficiency. Interestingly, in this study the expression of *StLBD1-5* homologous to *AtLBD37*, *AtLBD38*, and *AtLBD39* was also down-regulated during drought stress. This may affect the absorption by potato of nitrate because of the water shortage. We found that the color of potato leaf was deepened under drought (Figure 10), but there was no accumulation of purple matter visible to the naked eye. This shows that the low expression level of *StLBD1-5* may increase anthocyanin accumulation in the potato leaf to some degree. *StLBD1-5* was not the main factor causing anthocyanin accumulation, but the high expression of *StLBD1-5* was likely to strongly inhibit the accumulation of anthocyanins, thereby reducing the drought resistance of potatoes, which requires further testing. Similarly, *StLBD2-6* has a close relationship with *StLBD1-5*. They had similar effects on potato leaves. *StLBD2-6* and *StLBD3-5* up-regulated expression in the stem under drought stress, and they may have a certain protective effect when potato is under drought stress to maintain the normal physiological function of the stem. We hypothesize that *StLBD2-6* regulates a certain nitrogen metabolism pathway in the stem when potato is subjected to drought stress, in order to maintain normal physiological metabolism of the stem. The potato homologous genes *StLBD4-2*, *StLBD1-5*, *StLBD9-1,* and *StLBD9-2,* which were clustered together in Class II a, may have similar functions. Previous studies have shown that *AtLBD36* (*AtASL1*) is involved in the regulation of near-distal polarity and leaf morphogenesis [20,50]. Its corresponding potato homologs were *StLBD2-7*, *StLBD2-2,* and *StLBD2-1,* suggesting that these three homologous genes may also be involved in the regulation of flower development and leaf morphogenesis. 

Based on the homologous correspondence between the model plant *AtLBD* and the *StLBD*, we predicted the function of some *StLBD*. In the phylogenetic tree, *StLBD3-1* was in the same branch as *At-AS2* (*LBD6*), which was involved in flower development in *Arabidopsis* [18]. Similarly, according to the results of the heatmap, *StLBD3-1* was up-regulated in the tissue expression level heatmap of flowers (Figure 5). Therefore, it was very likely related to the development of potato flowers. Interestingly, *StLBD3-1* was also found to be scarcely expressed in the leaf, stem, tuber, and root of potato in our qRT-PCR results, which indicated that potato *StLBD3-1* might be similar to the *Arabidopsis At-AS2,* which was mainly expressed in the flower and involved in the regulation of potato flower development. Other researchers have found that the homologous gene *OsAS2* of *AtLBD6* in rice is involved in the regulation of rice bud differentiation and leaf formation [51]. Therefore, the function of *StLBD3-1* needs to be further verified. 

The overexpression of *Arabidopsis AtLBD41* in Celosia can cause leaf wrinkle deformity with significant distalization [17]. Therefore, we conjectured that the function of potato *StLBD3-5* was similar to that of *AtLBD41* and might be involved in the regulation of leaf near-far axis polarity. qRT-PCR results showed that the *StLBD3-5* expression level was up-regulated in the stem under drought stress (Figure 11C). The results showed that it displayed ABRE cis-acting in Figure 7, indicating that *StLBD3-5* had responded to ABA signaling and participated in polarity regulation. It has been reported that the overexpression of *AtLBD13* might significantly increase the formation of an elongated lateral root [52]. Based on the experimental results, qRT-PCR showed that the expression level of potato *StLBD6-5* homologous to *Arabidopsis AtLBD13* was very low in the root. This may be due to artificial error during sampling. The reason for this was that most of the slender lateral roots were pulled off and buried in the soil during sampling, so we only obtained the main root part. This also indicated that *StLBD6-5* is not expressed in the main root system of potato. What role it plays in the elongated lateral root of the potato remains to be further studied. Recent studies have shown that overexpression of the homologous gene *EgLBD29* of *Arabidopsis AtLBD13* in *Eucalyptus grandis Hill* will reduce the internode length and increase the degree of lignification. The homologous gene *EgLBD37,* which overexpresses *AtLBD1* and *AtLBD11,* enables plants to grow taller and grow between internodes [14]. Therefore, we hypothesize that overexpression of the potato *StLBD11-2* and homologous gene of *StLBD11-2* may make plants grow taller. Based on the homologous correspondence between the model plant *Arabidopsis AtLBD* gene and the potato *StLBD* gene, we predicted the function of some St*LBD* genes. In the phylogenetic tree (Figure 2), *StLBD3-1* and *At-AS2* (*LBD6*) were homologous genes, showing that StLBD3-1 might be involved in flower development [18]. *StLBD3-1* was shown in the tissue expression calorimetry which was highly expressed in flower that was likely to be related to the development of potato flowers. Potato homologous gene *StLBD4-2*, *StLBD1-5*, *StLBD9-1,* and *StLBD9-2* which were clustered with ClassII and may have similar functions of nitrate transport. Previous studies on *AtLBD36* (*AtASL1*) have shown that it is involved in regulation of the near-distal polarity and leaf morphogenesis of petals [20,50]. The relevant potato homologous genes were *StLBD2-7*, *StLBD2-2,* and *StLBD2-1.* These three potato homologous genes may also be involved in the control of flower development and leaf morphogenesis. *AtLBD16* (*AtASL18*), *AtLBD17* (*AtASL15*), *AtLBD18* (*AtASL20*) and *AtLBD29* (*AtASL16*) could induce callus culture formation and participate in plant regeneration. The above-mentioned corresponding potato genes *StLBD9-4*, *StLBD6-12*, *StLBD9 -3*, *StLBD1-4*, *StLBD6-3*, *StLBD1-2,* and *StLBD1-4* are also likely to be involved in similar regulatory pathways. *AtLBD16* and *AtLBD18* are both induced by auxin [53]. Their homologous gene *StLB1-4* was also induced by auxin IAA in the cluster analysis. According to the gene homology relationship, the function of the potato *LBD* gene above can be inferred, while the specific function is waiting to be confirmed by genetic transformation technology.

## 4. Materials and Methods 

### 4.1. Identification and Chromosomal Localization of Potato LBD Family Members

The complete protein sequence of potato was downloaded from potato genome sequencing consortium (PGSC) [28] (PGSC_DM_v3.4_pep. fasta. zip). The *Arabidopsis* LBD protein sequence was downloaded from the TAIR (https://www.arabidopsis.org/) database. Construction of a local BLAST database used potato whole genomic protein sequences. The software downloaded from ftp://ftp.ncbi.nlm.nih.gov/blast/executables/blast+/LATEST/. Local BLAST alignment using *Arabidopsis* LBD protein sequences (le-3) deleted the duplicate alignment results. The prefix “*St*” before the gene name indicated *Solanum tuberosum*. LBD indicated “Lateral organ boundaries domain”. Following numbers represented “Putative gene names” that were arranged by their top-down position on the chromosome. Letter “a” and letter “b” represented different transcripts (Table 1). The remaining protein sequences were subjected to multiple sequence alignment using Clustalx 1.83 [54]. Incomplete cod frame sequences and redundant sequences were manually deleted. Sequences containing conserved domains were screened. The selected sequences with strong conserved domains were submitted to the Pfam database (http://pfam.xfam.org/), and the proteins sequences containing the typical domain of LBD (DUF260, PF03195) were screened. The amino acid length, isoelectric point, and molecular weight of all identified potato LBD proteins were predicted using an online website (https://web.expasy.org/compute_pi/). Then, the basic information on the chromosome and position of the selected *LBD* gene on the chromosome from the PGSC was searched individually [28]. Chromosome localization was drawn using MapChart2.2 software [55]. 

### 4.2. Phylogenetic Tree Construction, Gene Structure Analysis and Protein Domain Sequence Alignment

Clustal W was used for multiple sequence alignment of potato LBD proteins sequences. The phylogenetic tree was constructed using the MEGA 7.0 software neighboring method (Neighbor-Joining, NJ), and the calibration parameter (Bootstrap) was set to 1000 times. The exon–intron structure information of the potato candidate *LBD* was predicted by the online software GSDS2.0 (http://gsds.cbi.pku.edu.cn/). A conserved domain in the potato LBD proteins sequence was predicted using the MEME tool (http://meme-suite.org/). 

### 4.3. Promoter Cis-Acting Element Analysis

The 1 kb nucleic acid sequence upstream of each candidate gene was manually searched for in the Phytozome database (https://phytozome.jgi.doe.gov/) and submitted to the promoter prediction database Plant Care (http://bioinformatice.psb.ugent.be/webtools/plantcare/html/), which predicts the promoter cis-acting element. In order to better understand these large super-secondary structures, we used EXCEL software to manually organize and construct the super-secondary motif cis-acting element expression heat map for each gene.

### 4.4. Tissue Expression and Induced Expression Analysis of the Potato LBD Gene

To explore the potential role of St*LBD* in growth and development, the transcriptome data of 16 tissues was downloaded from PGSC (http://solanaceae.plantbiology.msu.edu/index.shtml): flower, petiole, root, shoot apex, stamen, stem, stolon, tuber cortex, tuber peel, tuber pith, tuber sprout, leaf tuber (mature), tuber (young), and whole plant. An analysis of transcriptome data to the expression level (β-aminobutyric acid, methyl benzophenazole thioacetate, *Phytophthora infestans*, sodium chloride, mannitol, heat stress, abscisic acid, gibberellin, 6-benzylaminopurine, and auxin) of 10 stress treatments was conducted. Fragments per kilobase million (FPKM) values of each *LBD* gene were calculated, which indicated that their differentially expressed genes were identified. An expression heatmap of different tissues and different treatments of the LBD family was drawn by HemI.

### 4.5. Plant Material, Growth Conditions and Stress Treatment

Test material was the potato cultivar Desiree, which was planted in the greenhouse of the Northwest A&F University Laboratory, Yangling, China. The sampled po tato leaves had three biological replicates. The cut axillary buds were grown by tissue culture in MS medium (Murashige and Skoog, 1962) [41] containing 2% sucrose and 0.8% agar. The pH was 5.9 and maintained growth in an artificial climate greenhouse at 22 ± 1 °C 16 h light/8 h dark. Four-week-old seedlings were transferred to a vessel containing vermiculite and seedling substrate and were then watered. In order to ensure sufficient sampling material, two treatments (normal watering and drought stress) were set up for seven repetitions. Normal watering treatment included the provision of sufficient water every four days. The drought stress treatment included watering was only watered once on Day 0, after which the plants were no longer watered. Samples were taken every four days, which were quickly immersed in liquid nitrogen and frozen until the extraction of total RNA. On the last sampling of Day 20, the normal watered and drought-treated potato plants were dug out and divided into four parts: leaf, stem, tuber, and root. The collected material was quickly immersed in liquid nitrogen and frozen until the extraction of total RNA.

### 4.6. Total RNA Extraction and qRT-PCR Analysis

Total RNA from potato leaf was extracted using the RNAsimple Total RNA Extraction Kit (TIANGEN, Beijing, China, DP419). cDNA was synthesized using the FastKing RT Kit (With gDNase) (TIANGEN, Beijing, China). The cDNA was diluted five-fold with nuclease-free water, and all procedures were performed according to the instructions. The specific potato *LBD* gene primers for quantitative real-time PCR (qRT-PCR) were designed with Primer Premier 5 (Table 2). qRT-PCR was carried out in a 10 μL reaction system with the following composition: 5 µL 2 × SuperReal Color PreMix, 0.5 µL 10 µM forward primer, 0.5 µL 10 µM reverse primer, 1 µL diluted cDNA and 3 µL ddH_2_O. The qRT-PCR program was set to: 15 min at 95 °C, 10 s at 95 °C, 20 s at 63 °C, and 30 s at 72 °C. The dissolution profile was generated from 65 °C to 95 °C and increased by 0.5 °C every 5 s for three biological replicates. The relative expression was calculated using the 2^−ΔCT^ method of the tissue expression profile and 2^−ΔCT^ method of expression profiles under stress [5]. The reference gene ubi3 was used as a qRT-PCR reference gene (Table 2). The qRT-PCR raw data can be found in Appendix A. Expression levels were analyzed using a Bio-Rad Real-Quantitative real-time PCR analysis System (CFX96, Hercules, CA, USA).

## 5. Conclusions

In this study, a comprehensive analysis on the identification of potato LBD family members was carried out by bioinformatics. Forty-three *StLBD* family genes were obtained in total. Except for chromosomes 7 and 10, the other 10 chromosomes all had the *LBD* genes. We found that 41 of 43 genes were expressed in at least one tissue. Some genes expression levels were very high in specific tissues, such as *StLBD1-5*, *StLBD2-6*, *StLBD3-5,* and *StLBD11-1*. Induced expression analysis showed that the potato *LBD* gene mainly responded to immune, drought, and heat stress. However, there was no obvious gene expression change in ABA stress. Combining qRT-PCR experiments with previous studies, we believed that potato *StLBD1-5*, *St-LBD2-6*, *StLBD3-1*, *StLBD6-5,* and *StLBD11-2* have important research value. A low expression of *StLBD1-5* and *StLBD2-6* under drought stress was shown to be beneficial for increasing the drought resistance of the leaf. The high expression of *StLBD3-5* and *StLBD2-6* in the stem was revealed to be beneficial for increasing the ability of the potato to resist drought stress. In the early stage of the *StLBD3-1* response to drought stress, a low expression of *StLBD6-5* could improve the drought resistance of potato leaf. *StLBD11-2* was not affected by drought stress, while it displayed periodicity change with leaf development. In subsequent studies, we will try to reveal StLBD transcription factors’ response to drought stress and the biological process of StLBD transcription factors under drought stress. We will enhance drought tolerance through the regulation of *StLBD* in potatoes. 

## Figures and Tables

**Figure 1 ijms-20-05360-f001:**
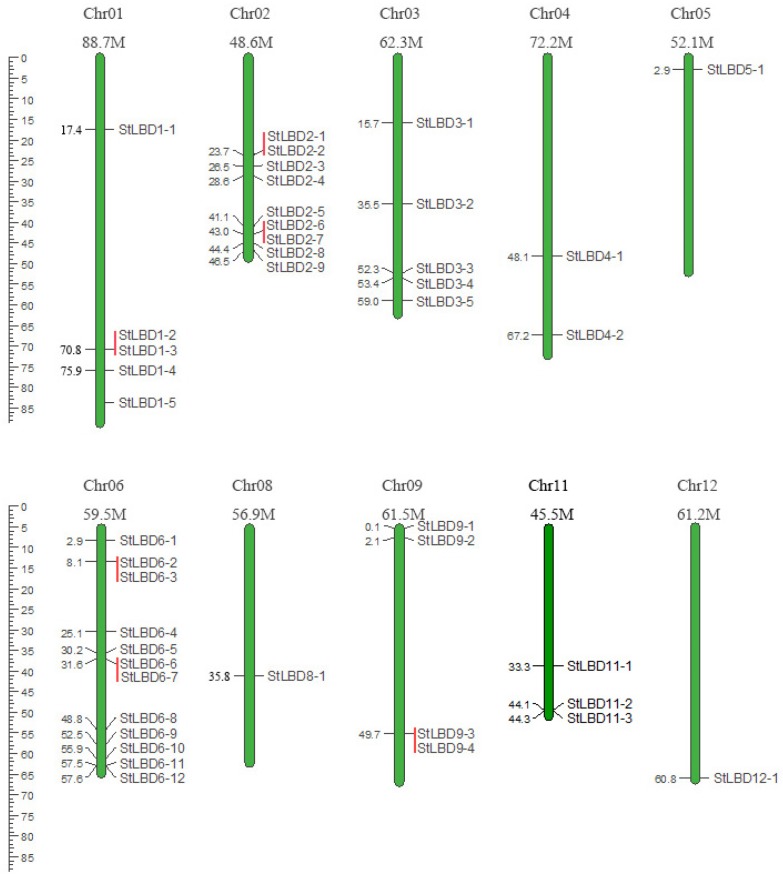
The location of the Lateral organ boundaries domain (*LBD*) genes in potato on the chromosome. A basic unit represents a chromosome length of 5.0 Mb. The number of each chromosome is marked above it.

**Figure 2 ijms-20-05360-f002:**
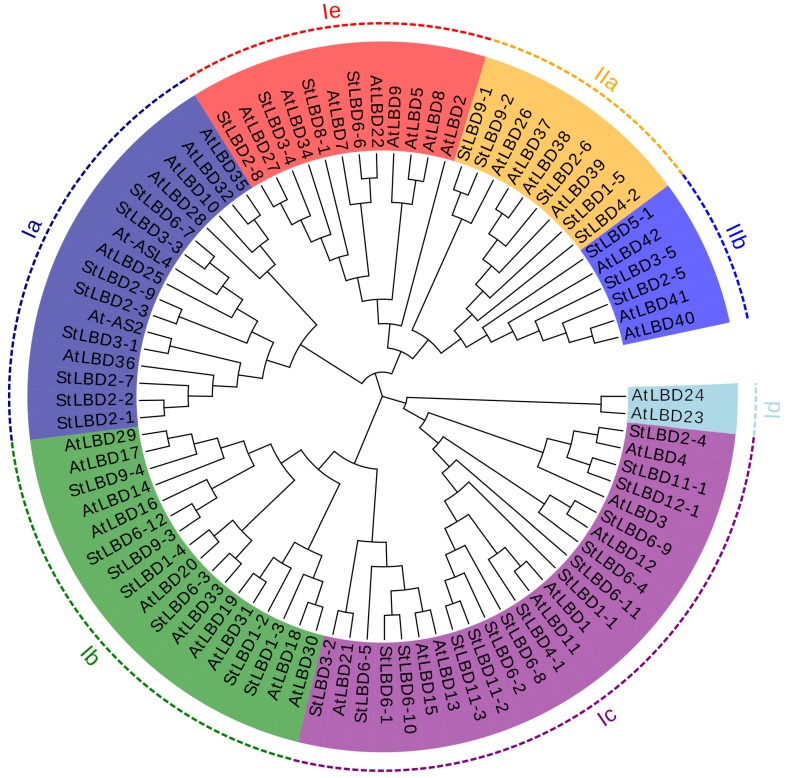
The phylogenetic tree of LBD transcription factors of potato and *Arabidopsis*. The clustering analysis is based on 1000 replications for increasing the credibility of the bootstrap value.

**Figure 3 ijms-20-05360-f003:**
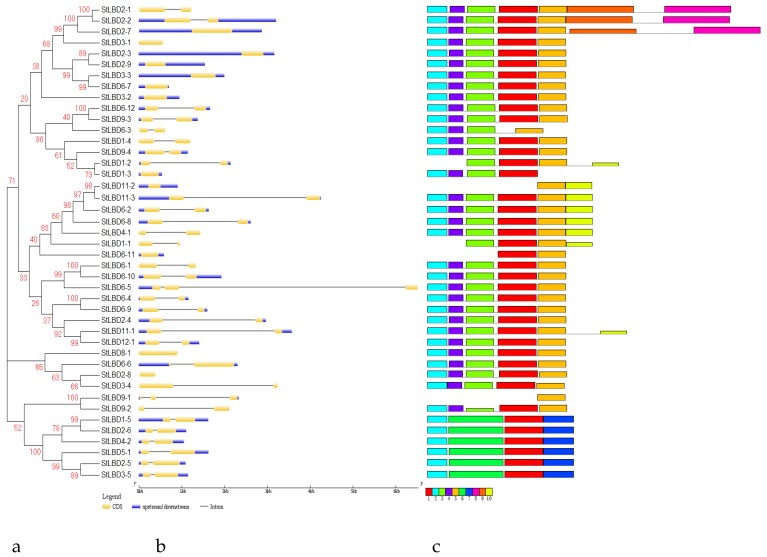
Phylogenetic relationships of the potato *LBD* gene family (**a**), gene structure (**b**) and conserved domain (**c**). The potato LBD amino acid sequence was aligned by Clustal X 1.83. The phylogenetic tree was constructed using the N-J method of MEGA7. Bootstrap verification produced 1000 duplicate bootstrap values. These genomic sequences corresponding to their cod sequences (CDS) were predicted online using GSDS (http://gsds.cbi.pku.edu.cn) to detect the intron/exon distribution of the corresponding *StLBD* gene. The yellow boxes indicate the cod sequences (CDS). The discontinuous lines indicate the introns of these genes. The right side represents the motif composition associated with each StLBD protein. The motifs, numbered 1-10, are displayed in different colored boxes.

**Figure 4 ijms-20-05360-f004:**
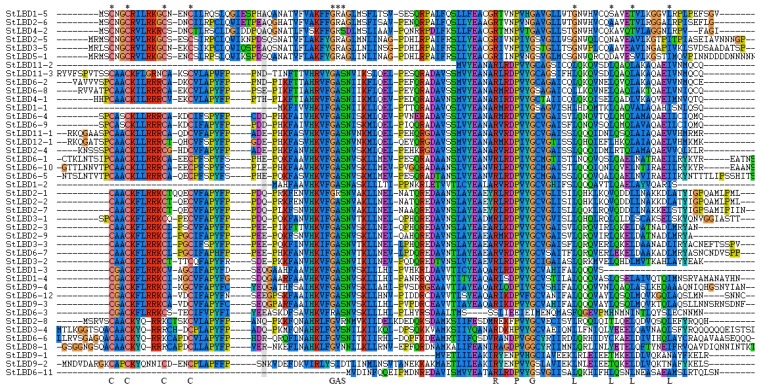
Potato LBD family conserved domain protein sequence alignment. The amino acid sequence of the potato LBD protein was aligned by Clustal X 1.83. “*” Indicates the CX2CX6CX3 zinc finger-like motif, GAS (Gly-Ala-Ser) block, and LX6LX3LX6L leucine zipper-like coiled-coil motif [14].

**Figure 5 ijms-20-05360-f005:**
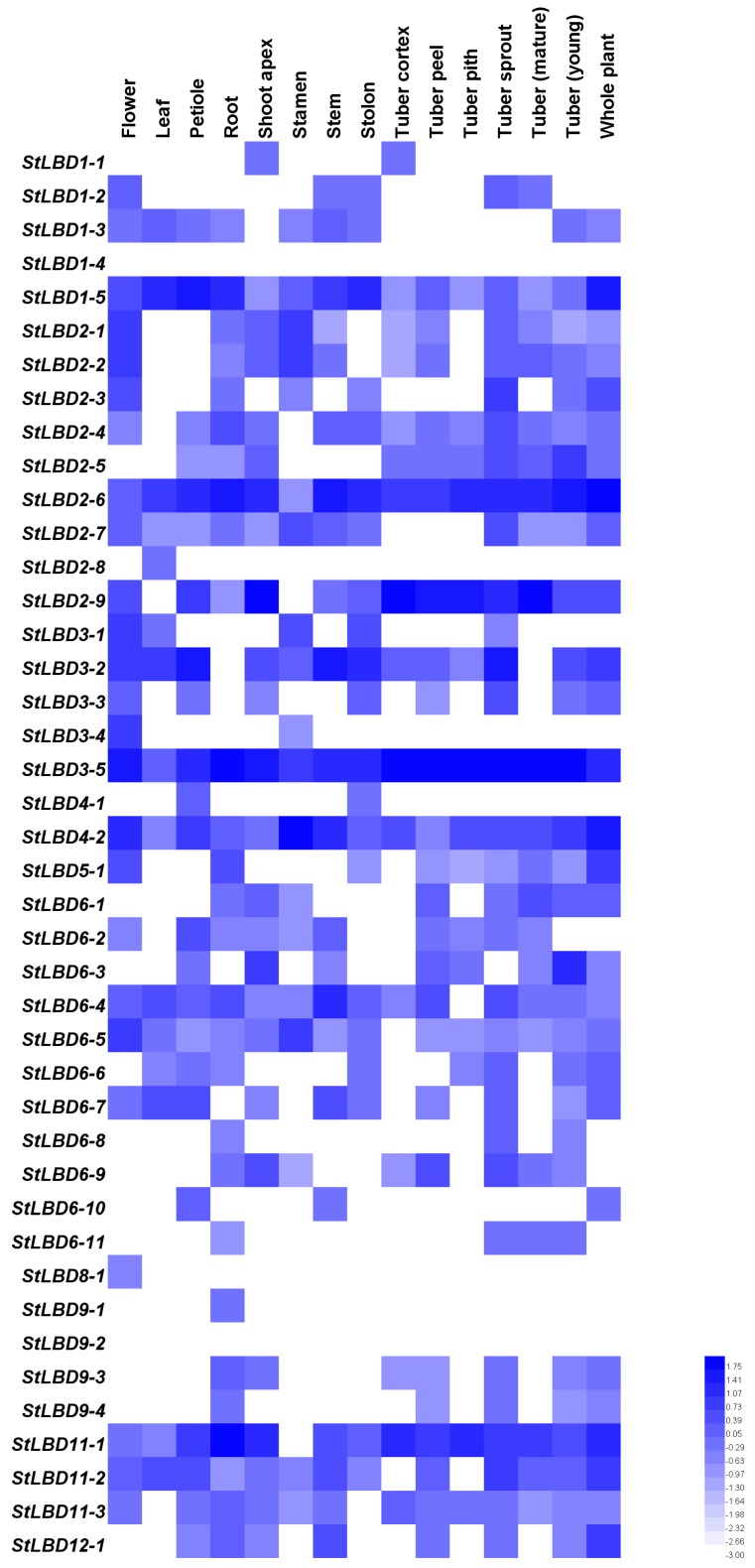
The expression profiles of *StLBD* genes in different tissues of potato. Transcripts were detected by RNA-Seq technology. The heatmap showed *StLBD* gene expression across 15 tissues covering the entire potato life cycle, including the flower, leaf, petiole, and root, shoot apex stamen stem stolon, tuber cortex, tuber peel, tuber pith, tuber sprout, mature tuber, young tuber, and whole plant. In this figure, blue and white are used to indicate the expression level of each tissue gene. Blue indicates gene up-regulation, while white indicates gene down-regulation.

**Figure 6 ijms-20-05360-f006:**
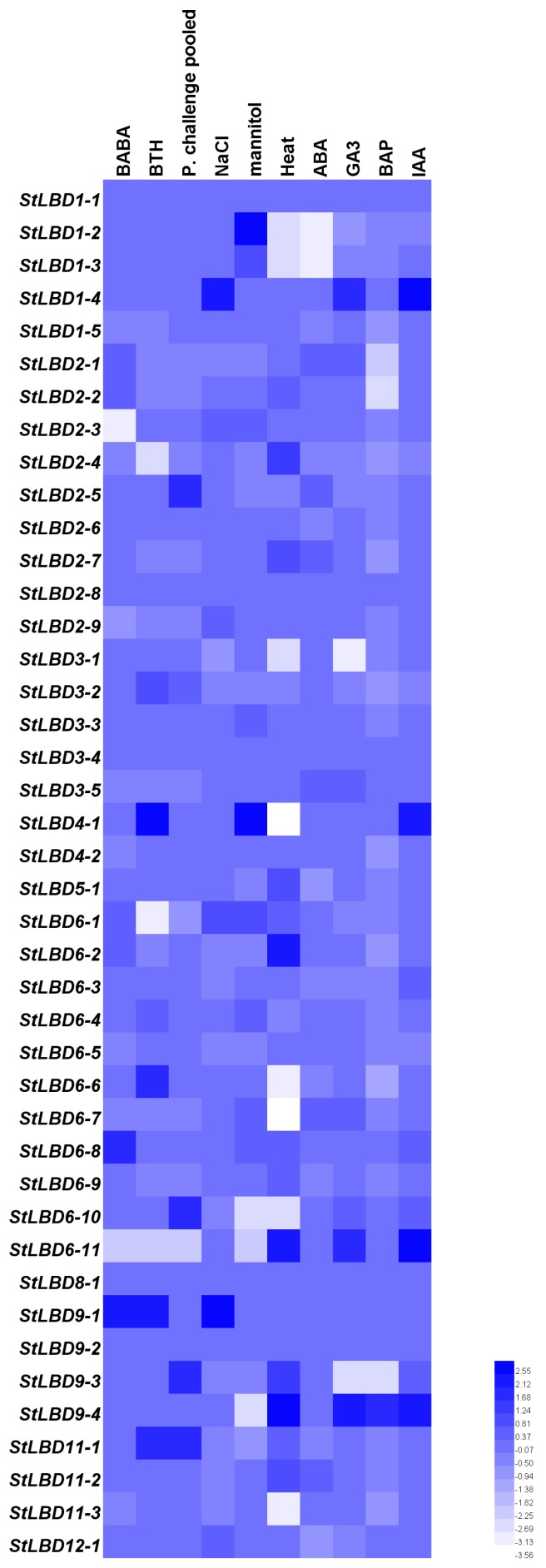
Heatmap of the expression profile of potato *LBD* genes under ten different biotic or abiotic stresses. Transcripts were detected with RNA-Seq technology. Abiotic stresses include sodium chloride, mannitol, and heat. Biological stresses include *Phytophthora infestans*, and stress-elicitors acibenzolar-S-methyl (BTH) and DL-b-amino-n-butyric acid (BABA). Other stress responses were mainly induced by four plant hormones: abscisic acid (ABA), 6-benzylaminopurine (BAP), gibberellic acid (GA3), and indole-3-acetic acid (IAA). Blue and white in the figure were used to indicate the gene expression level in plant tissues under specific conditions. Blue indicates gene up-regulation, while white indicates gene down-regulation.

**Figure 7 ijms-20-05360-f007:**
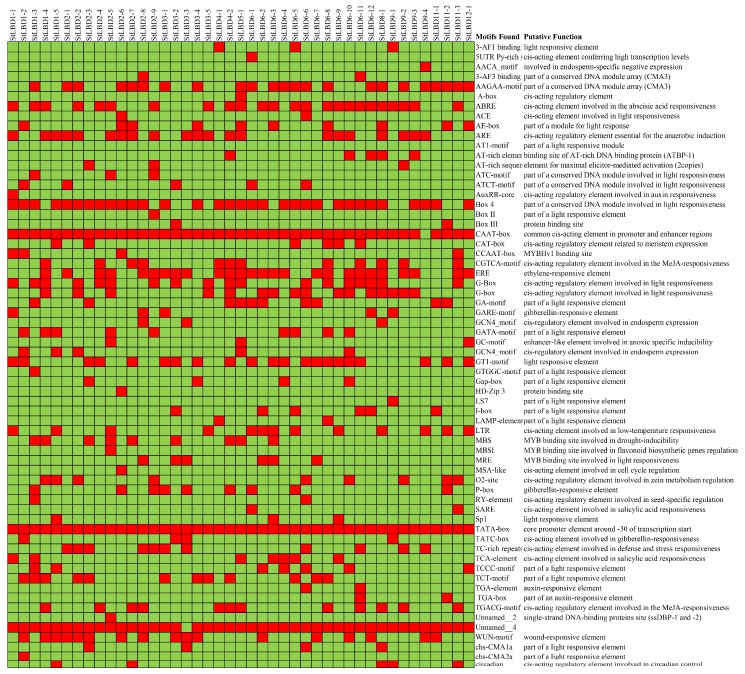
The cis-acting element of 1000 bp sequences upstream of the potato *StLBD* gene. This study used the database PlantCARE to predict the motif. Red indicates the existence of a specific motif, and green indicates that no specific motif is present.

**Figure 8 ijms-20-05360-f008:**
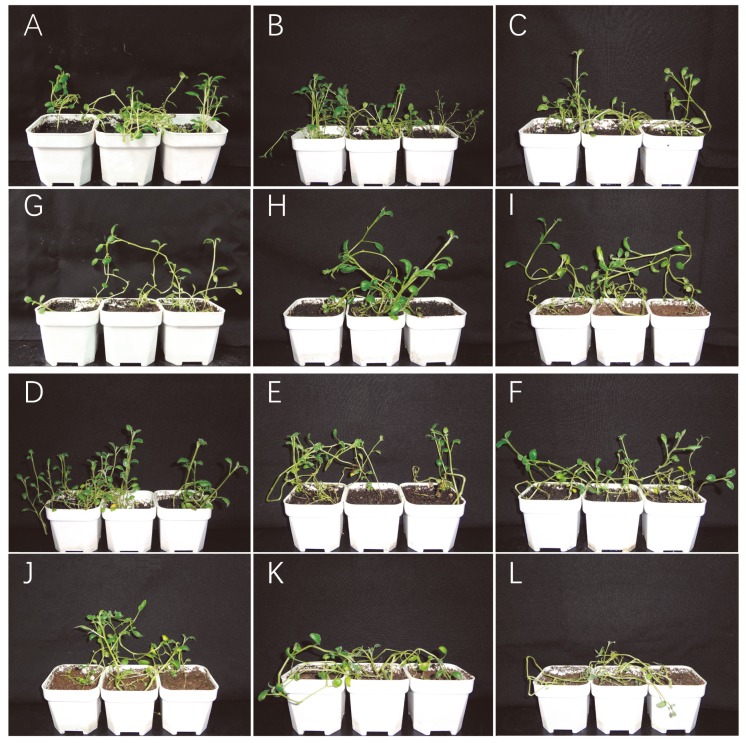
Potato seedlings treated with drought stress and watering in six periods. (**A**–**F**) are used to represent the variety of Desiree seedlings with watering for 0 days, 4 days, 8 days, 12 days, 16 days and 20 days, respectively. (**G**–**L**) to represent Desiree seedlings of potato varieties treated with drought stress for 0 days, 4 days, 8 days, 12 days, 16 days, and 20 days, respectively.

**Figure 9 ijms-20-05360-f009:**
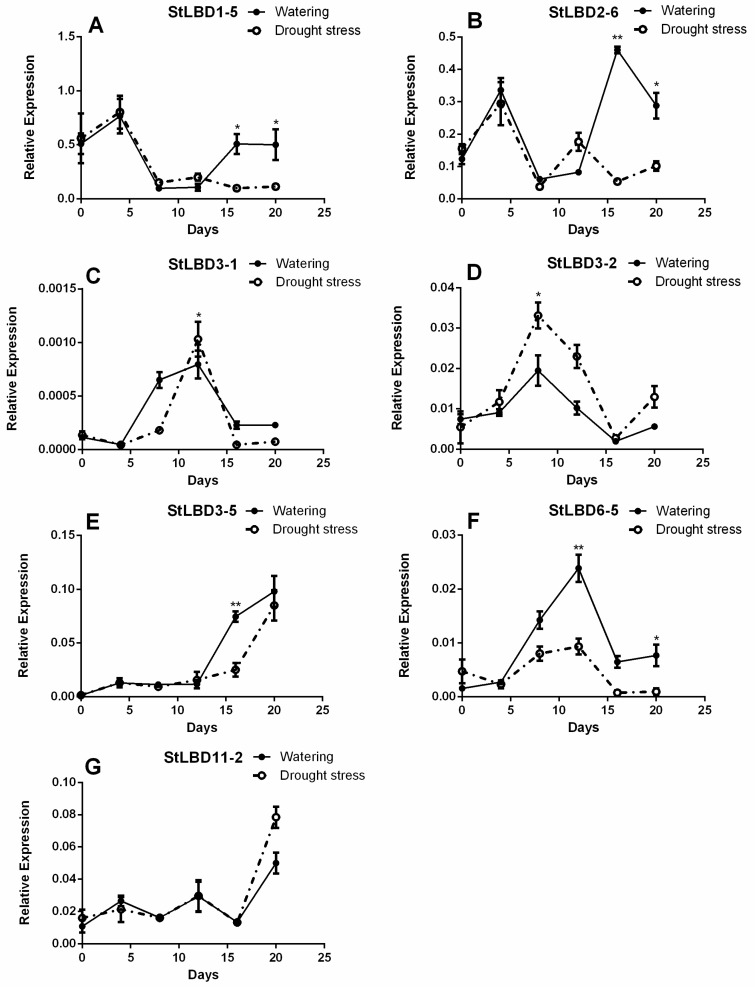
Curve of expression of the StLBD gene with time under drought stress and normal watering treatment. Asterisks indicate significant differences using Student’s *t* test (* *p* < 0.05; ** *p* < 0.01).

**Figure 10 ijms-20-05360-f010:**
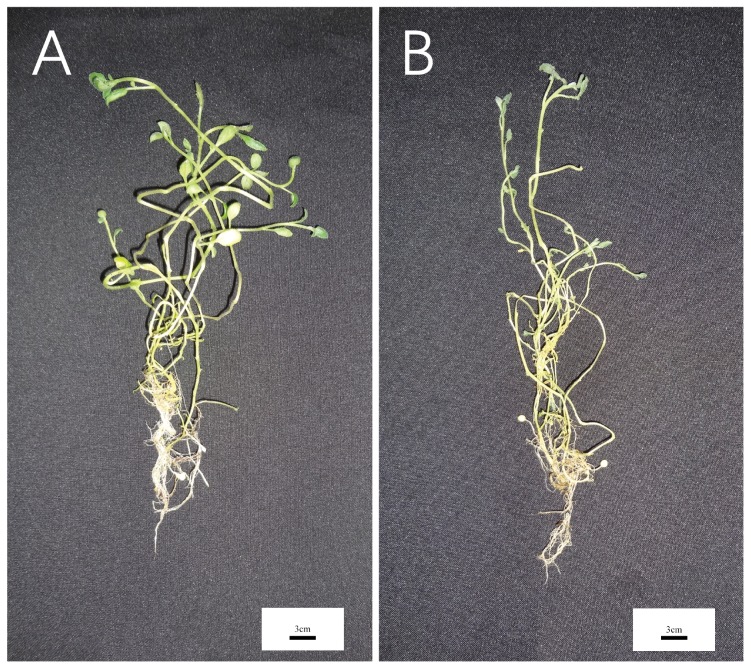
Potato seedlings treated with normal watering and drought stress for 20 days. (**A**) Desiree seedlings of potato varieties treated with normal watering for 20 days. (**B**) Desiree seedlings of potato variety treated for 20 days under drought stress.

**Figure 11 ijms-20-05360-f011:**
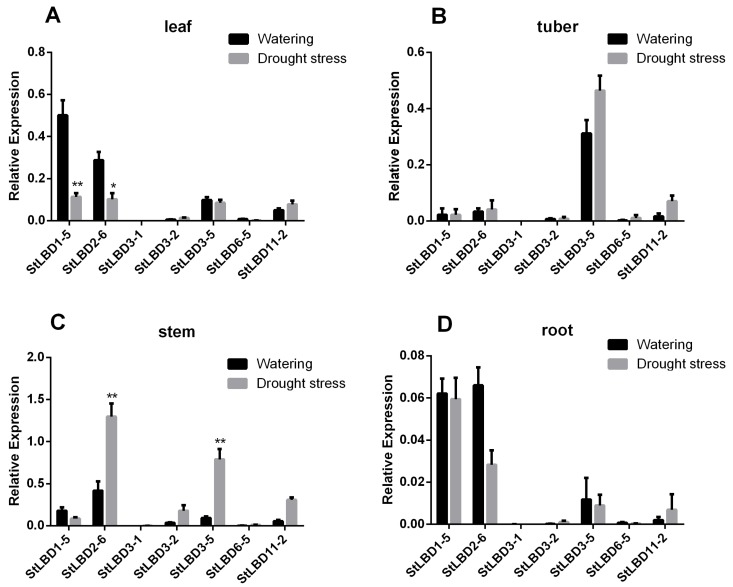
Relative expression level of the *StLBD* gene in the leaf, tuber, stem, and root under drought stress and normal watering treatment. Asterisks indicate significant differences using Student’s *t* test (* *p* < 0.05; ** *p* < 0.01).

**Table 1 ijms-20-05360-t001:** Information on the lateral organ boundaries domain (*LBD*) transcription factor family in *Solanum tuberosum*.

Gene Features Phytozome Gene ID	Phytozome Transcript ID	Protein Features								
		Families	Putative Gene Name	Exon No.	Chr. Location	Protein Length	MW (kDa)	*pI*	GRAVY	TMD/Terminus
PGSC0003DMG400014757	PGSC0003DMT400038248*	Ic	*StLBD1-1*	2	chr01	128	15	5.53	−0.293	0/out→out
PGSC0003DMG400012757	PGSC0003DMT400033221*	Ib	*StLBD1-2*	2	chr01	133	15	6	0.187	0/out→out
PGSC0003DMG402012772	PGSC0003DMT400033253*	Ib	*StLBD1-3*	1	chr01	139	14	9.12	0.136	0/out→out
PGSC0003DMG400022454	PGSC0003DMT400057829*	Ib	*StLBD1-4*	2	chr01	241	26	6.13	−0.393	0/out→out
PGSC0003DMG400025752	PGSC0003DMT400066165*	IIa	*StLBD1-5*	2	chr01	217	24	6.9	−0.267	0/out→out
PGSC0003DMG401000764	PGSC0003DMT400002011*	Ia	*StLBD2-1*	2	chr02	288	33	6.44	−0.839	0/out→out
PGSC0003DMG402000764	PGSC0003DMT400002013*	Ia	*StLBD2-2*	2	chr02	287	33	6.23	−0.806	0/out→out
PGSC0003DMG400006914	PGSC0003DMT400017806*	Ia	*StLBD2-3*	1	chr02	168	19	8.89	−0.649	0/out→out
PGSC0003DMG400021204	PGSC0003DMT400054637*	Ic	*StLBD2-4*	2	chr02	161	18	8.21	−0.442	0/out→out
PGSC0003DMG400012653	PGSC0003DMT400032938*	IIa	*StLBD2-5*	2	chr02	273	30	7.02	−0.392	0/out→out
PGSC0003DMG400024936	PGSC0003DMT400064185*	IIa	*StLBD2-6a*	2	chr02	212	23	6.11	−0.231	0/out→out
	PGSC0003DMT400064186	IIa	*StLBD2-6b*	2	chr02					
PGSC0003DMG400010023	PGSC0003DMT400025959*	Ia	*StLBD2-7*	1	chr02	313	35	5.98	−0.641	0/out→out
PGSC0003DMG400038771	PGSC0003DMT400089200*	Ie	*StLBD2-8*	1	chr02	123	15	8.93	−0.658	0/out→out
PGSC0003DMG400012629	PGSC0003DMT400032877*	Ia	*StLBD2-9*	1	chr02	157	18	5.91	−0.719	0/out→out
PGSC0003DMG400000974	PGSC0003DMT400002549*	Ia	*StLBD3-1*	1	chr03	189	21	8.77	−0.529	0/out→out
PGSC0003DMG400027718	PGSC0003DMT400071281*	Ic	*StLBD3-2*	1	chr03	177	20	6.29	−0.388	0/out→out
PGSC0003DMG400018112	PGSC0003DMT400046644*	Ia	*StLBD3-3*	1	chr03	191	21	8.21	−0.469	0/out→out
PGSC0003DMG400013182	PGSC0003DMT400034288*	Ie	*StLBD3-4*	2	chr03	301	34	5.48	−0.579	0/out→out
PGSC0003DMG400005719	PGSC0003DMT400014630*	IIa	*StLBD3-5*	2	chr03	244	27	6.44	−0.361	0/out→out
PGSC0003DMG400004138	PGSC0003DMT400010597*	Ic	*StLBD4-1*	2	chr04	173	20	5.71	−0.467	0/out→out
PGSC0003DMG400021509	PGSC0003DMT400055393*	IIa	*StLBD4-2a*	2	chr04	204	22	5.07	−0.208	0/out→out
	PGSC0003DMT400055392	IIa	*StLBD4-2b*	2	chr04					
PGSC0003DMG400014547	PGSC0003DMT400037711*	IIa	*StLBD5-1*	2	chr05	253	28	5.47	−0.426	0/out→out
PGSC0003DMG400020562	PGSC0003DMT400052995*	Ic	*StLBD6-1*	2	chr06	201	22	6.81	−0.209	0/out→out
PGSC0003DMG400025400	PGSC0003DMT400065346*	Ic	*StLBD6-2a*	2	chr06	209	23	5.16	−0.273	0/out→out
	PGSC0003DMT400065345	Ic	*StLBD6-2b*	1	chr06					
PGSC0003DMG400046838	PGSC0003DMT400097267*	Ib	*StLBD6-3*	2	chr06	155	18	5.4	−0.051	0/out→out
PGSC0003DMG400023939	PGSC0003DMT400061511*	Ic	*StLBD6-4*	2	chr06	171	19	5.73	−0.18	0/out→out
PGSC0003DMG400029080	PGSC0003DMT400074783*	Ic	*StLBD6-5a*	3	chr06	274	30	4.98	−0.469	0/out→out
	PGSC0003DMT400074784	Ic	*StLBD6-5b*	2	chr06					
PGSC0003DMG400004880	PGSC0003DMT400012498*	Ie	*StLBD6-6*	2	chr06	327	37	5.48	−0.895	0/out→out
PGSC0003DMG400027078	PGSC0003DMT400069646*	Ia	*StLBD6-7*	1	chr06	175	19	6.29	−0.485	0/out→out
PGSC0003DMG400007503	PGSC0003DMT400019410*	Ic	*StLBD6-8*	2	chr06	198	22	6.71	−0.282	0/out→out
PGSC0003DMG400030456	PGSC0003DMT400078256*	Ic	*StLBD6-9*	2	chr06	173	19	6.7	−0.221	0/out→out
PGSC0003DMG400030461	PGSC0003DMT400078265*	Ic	*StLBD6-10a*	2	chr06	222	24	9.02	−0.209	0/out→out
	PGSC0003DMT400078264	Ic	*StLBD6-10b*	2	chr06					
PGSC0003DMG400027663	PGSC0003DMT400071132*	Ic	*StLBD6-11*	1	chr06	134	15	4.87	−0.35	0/out→out
PGSC0003DMG400020069	PGSC0003DMT400051680*	Ib	*StLBD6-12*	2	chr06	195	22	6.49	−0.341	0/out→out
PGSC0003DMG400044009	PGSC0003DMT400094438*	Ie	*StLBD8-1*	1	chr08	302	33	8.85	−0.441	0/out→out
PGSC0003DMG400008500	PGSC0003DMT400021914*	IIa	*StLBD9-1*	3	chr09	108	12	5.39	−0.81	0/out→out
PGSC0003DMG400043103	PGSC0003DMT400093532*	IIa	*StLBD9-2*	2	chr09	167	19	4.9	−0.559	0/out→out
PGSC0003DMG400009713	PGSC0003DMT400025135*	Ib	*StLBD9-3*	2	chr09	221	25	8.68	−0.484	0/out→out
PGSC0003DMG400009716	PGSC0003DMT400025142*	Ib	*StLBD9-4*	2	chr09	245	28	5.66	−0.514	0/out→out
PGSC0003DMG400008649	PGSC0003DMT400022286*	Ic	*StLBD11-3*	2	chr11	171	19	6.94	−0.371	0/out→out
PGSC0003DMG400027372	PGSC0003DMT400070405*	Ic	*StLBD11-1*	1	chr11	94	11	4.16	−0.264	0/out→out
PGSC0003DMG400025384	PGSC0003DMT400065297*	Ic	*StLBD11-2*	2	chr11	216	23	5.05	−0.219	0/out→out
PGSC0003DMG402023851	PGSC0003DMT400061283*	Ic	*StLBD12-1*	2	chr12	169	19	9.01	−0.295	0/out→out

a. The number of transmembrane domains is represented by TMD. Terminus shows the predicted location of N- and C-terminals, OUT (extra-cellular). Arrow (→) shows from N- to C-terminals. b. “*” Primary transcript of a specified gene. c. The “MW” is molecular weight. The “pI” is isoelectric point. The “GRAVY” is grand average of hydropathicity. The TMD is transmembrane domain.

**Table 2 ijms-20-05360-t002:** qRT-PCR primers for expression in the analysis of *StLBD*.

Gene	Forward Primer (5′-3′)	Reverse Primer (5′-3′)
*Ubi3*	TCCGACACCATCGACAATGT	CGACCATCCTCAAGCTGCTT
*StLBD1-5*	CAATGCCACTGTCTTCGTCG	CTCCTTTAAGCACTGTTTCTACCG
*StLBD2-6*	CGAACAGTAAACCCAGTGAACG	TGATCGAGATCCGCCAACAT
*StLBD3-1*	CGGTTCCTTCGTCGTAAATGTC	GCACGCATGTCTGCCTCATA
*StLBD3-2*	CAACGAGGCTGAGGTGAGACTTA	TGACCAGAAGTGTCCGCAAAA
*StLBD3-5*	CTTGGACCTGACCATTTGCG	GGGAGATCGGGATCAGAGTTATTT
*StLBD6-5*	CCTGAATGTTATTAGAGCGGAGAT	CGAGGTGGTTGTAGACGGTTG
*StLBD11-2*	CGCTAAGGCACAAGCTGAAA	ATAGGGTCTCCCATGATCCAAT

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
