# Peer review of "Genome-Wide Analysis of the Lateral Organ Boundaries Domain (LBD) Gene Family in Solanum tuberosum"

_ijms, 2019, doi:10.3390/ijms20215360_

Round 1

Reviewer 1 Report

The authors have conducted a novel and interesting study on LBD proteins, and the regulatory responses to drought stress. This is important work on a critical food crop, but the presentation of the study is significantly marred by a poorly written manuscript. The text is full of sentence fragments, inappropriate mid-sentence capitalisation and misused punctuation. Genus and species names must be italicised, in both the paper title and the text.

The science here is sound, but the paper needs significant attention before it can be published.

Author Response

Response to Reviewer 1 Comments

Declaration: All changes in the manuscript are marked in red.

Point: .

The authors have conducted a novel and interesting study on LBD proteins, and the regulatory responses to drought stress. This is important work on a critical food crop, but the presentation of the study is significantly marred by a poorly written manuscript. The text is full of sentence fragments, inappropriate mid-sentence capitalisation and misused punctuation. Genus and species names must be italicised, in both the paper title and the text.

The science here is sound, but the paper needs significant attention before it can be published.

Response:.

Thank you for your interest in our research and suggestions for changes.

We have already spent money to polish the manuscript. However, the result is not very good. The level of this language service company is not good. We hope that you can give opportunities and we will once again modify the manuscript until the manuscript is received.

We have corrected some grammar errors and corrected inappropriate mid-sentence capitalisation and misused punctuation. If there are still errors in the manuscript, we will modify it again.

We have changed genus and species names to italics, in manuscript both the paper title and the text. All changes are marked in red.

This time we made the following changes:

Original manuscript L18-L19:”However, potato LBD protein has not been studied.” changed to L19-L20:“However, the potato systematics and evolution of LBD gene family have not been fully conducted by scholars and researchers.”

Original L387-l388 “but the identification of the potato LBD gene family at the genomic level has not been reported yet” changed L391-L392“However, the identification of the potato LBD gene family at the genomic level has not been fully reported.”

We referenced conclusions in the introduction by Zebarth, B.J, et al. in 2012: L75-L77 ” Zebarth, B.J, et al. find that the gene St.LBD (Unigene Stu.5076 from www.cpgp.ca) is homologous to AtLBD37, AtLBD38 and AtLBD39 in potato may be specifically in response to nitrate rather than N deficiency[27]”

We referenced the conclusions in the Result section that by Bdeir R et al. (DOI https://doi.org/10.1007/s00606-015-12) and discuss L149-L153 ”Interestingly, our results are roughly the same as those of the incomplete potato study by Bdeir R et al[28]. The difference is we have found 7 homologous genes of potato by AtLBD1 which 5 have higher similarities: StLBD4-1, StLBD6-8, StLBD6-2, StLBD11-2, StLBD11-3. 2 similarities are low: StLBD6-11, StLBD1-1. We also find a homologous gene of AtLBD18, it is StLBD1-3, that has not been found by predecessors.”

L315-L317 changed to:” Over time the expression level of StLBD2-6 is similar to StLBD1-5. After 16 days, these two treatments begin to show significant differences and down-regulated expression.”

L341-L342 increased:” The physiological state is shown in Figure 11.”

L348-L349 changed to:” but still retains a certain amount of water has cell viability and has grown tubers (Figure 11B).”

We deleted Original manuscript L419-L421:” Such as StLBD2-6 and StLBD3-5 in the stem, they may have a certain protective effect when the potato is under drought stress to maintain the normal physiological function of the stem.”

We incareased disscution in the Disscution section and modified some sentences L417-L443:” However, the expression level of StLBD3-2 and StLBD11-2 are lower than has been reported transcript data in the leaves which in this study. We found that the expression levels of LBD genes had significant difference in different tissues. The expression of some genes show significant differences under the arid condition. Anthocyanins play a key role in the plant defence [49]. Previous studies, AtLBD37, AtLBD38 and AtLBD39 are negative regulators of anthocyanin synthesis. Overexpression of each of the three genes in the absence of N/NO3- strongly suppresses the key regulators of anthocyanin synthesis [50]. Therefore, we conclude that high expression of these LBD genes is detrimental to increasing plant stress tolerance. DFR gene is an enzyme in the anthocyanin biosynthesis pathway [51]. With the supply of high nitrates, DFR expression is reduced, anthocyanin accumulation is reduced by Zebarth, B.J. et al. This is associated with an increased expression of the N-regulated anthocyanin synthesis inhibitor St.LBD (Unigene Stu. 5076 from www.cpgp.ca). However, ammonium increases has no effect on the expression of St.LBD. Prove that the gene may function specifically in response to nitrate rather than N deficiency. Interestingly, in this study the expression of StLBD1-5 that homologous to AtLBD37, AtLBD38 and AtLBD39 was also down-regulated during drought stress. This may be affecting the absorption of potato to nitrate because of water shortage. We found that the color of potato leaves is deepened under drought (Figure 11) but there is no accumulation of purple matter visible to the naked eye. This shows that the low expression level of StLBD1-5 may increase anthocyanin accumulation in potato leaf to some degree. StLBD1-5 is not the main factor causing anthocyanin accumulation but the high expression of StLBD1-5 is likely to strongly inhibit the accumulation of anthocyanins, thereby reducing the drought resistance of potatoes, which requires further testing. Similarly, StLBD2-6 which has a close relationship with StLBD1-5, they are similar roles in the leaf of potato. The difference is, StLBD2-6 and StLBD3-5 up-regulated expression in stems under drought stress, they may have a certain protective effect when potato under the drought stress to maintain the normal physiological function of the stem. We presume that StLBD2-6 regulates a certain nitrogen metabolism pathway in the stem when potato is subjected to drought stress, in order to maintain the normal physiological metabolism of the stem.”

We deleted original manuscript L421-L424:” It is noteworthy that the potato StLBD2-6 and StLBD1-5 genes are highly homologous to Arabidopsis AtLBD37, AtLBD38 and AtLBD39. Studies have shown that AtLBD37 (AtASL39), AtLBD38 (AtASL40) and AtLBD39 (AtASL41) genes are involved in anthocyanin synthesis and nitrogen metabolism in Arabidopsis”

We deleted original manuscript L427-L430:” Similarly, the transcriptome heat map showed that the potato LBD2-6 gene had the highest expression relative to other potato LBD genes, so it is likely to be involved in the regulation of nitrogen transport in potato plants.”

We deleted original manuscript L469-L471:” Studies have also shown that AtLBD37 (AtASL39), AtLBD38 (AtASL40) and AtLBD39 (AtASL41) genes are involved in anthocyanin synthesis and nitrogen metabolism in Arabidopsis”

original manuscript : “may have similar functions.””changed to new manuscript L482” similar function of nitrate transport.”

We have changed the name of all Arabidopsis to italics, such as: L95 ” Arabidopsis”. The names of the potatoes are all changed to italics such as L86:” Solanum tuberosum”. Other than this, L43:” Oryza sativa, Malus domestica and Eucalyptus grandis Hill”, L258-L259: ” P. infestans” etc. all have been changed to italics. We have marked them all in red.

We have changed L69 "MdLBD13" in the " Apple MdLBD13 gene can inhibit anthocyanin synthesis and nitrogen uptake " to italic " MdLBD13 " and carefully checked the whole the manuscript, corrected all such errors. Such as L59: “AtLBD41 (AtASL38)”,L65 “OsAS2”, L71” EgLBD37” etc. We changed all gene names in Table 1 and Table 2 to italics that have been marked in red.

We have changed in original manuscript L42-L43: ”giant salamander” to in new manuscript L43: “Eucalyptus grandis Hill

We have deleted L125-L126 (original manuscript L124-L125): "We know that" from the manuscript changed that to “Analysis of the chromosomal localization results from the potato LBD gene (Figrue 1), 43 LBD genes is distributed into 10 of the 12 chromosomes.”

We changed the color of the expression level heatmap (figure 5 and figure 6) to blue and white (original is red-black-green). Blue represents up-regulation of expression level and white represents down-regulation of expression level. If you have a better suggestion about the color of the heatmap please tell us right away and we will modify it in time.

We put the figure 10 picture G below A, H below B and so on.

We incareased L312-L313: ” On the 16th day, the potato begins to show significant differences under the arid condition (Figure 10K).”

We changed L318-L320 to:” The expression level of StLBD3-1 is very low in leaf, the expression level of StLBD3-1 is the highest on the 12th day under the arid condition and seedling is no lodging at this time (Figure 10J).”

We have correced H2O in L560(original L553) into “H2O”.

We deleted repeated references and ensured that the references are correct.

L239-L244: Change the unit "uM" of to "μM"

We changed the gene names to italics, in both the figure 5 and figure 6.

L202: changed “red-black-green” to “blue and white”, changed” red” to “blue” and so on.

L202: changed “green” to “white” and L254: changed “green” to “white” and so on.

We changed Figure 9 for the following reasons:

First declare that the conclusions we have reached are correct.

In the Figure 9 of original manuscript, we have carried out data conversion that in order to beauty of the picture. However, such show is easy to mislead the reader. The reason is that the expression level of StLBD3-1 is very low, and error caused by the test is higher than the difference. The expression level of StLBD3-1 is very low in any tissues, and there is no such difference in the original picture. So we re-uploaded the figure 9.

Reviewer 2 Report

The main remarks concern the statement: “there have been no reports on the potato LBD gene family”. However short literature search give me at least 2 results, which should be mention in this work: the paper by Bernie et al. 2012 (DOI https://doi.org/10.1007/s12230-012-92) published in American Journal of Potato Research and the paper by Bdeir et al. 2016 (DOI https://doi.org/10.1007/s00606-015-12) published in Plant Systematics and Evolution. However they are not so comprehensive look into Solanaceae LBD gene family and possible function of their products, but both of them should be mention. You need to change whole manuscript accordingly.

Authors used Arabidopsis as a name of Arabidospsis thaliana L. The name written with upper case should be written italics as it is Latin name. If the arabidopsis is used as English name should be written with lower case, however the proper English name is the thale cress. Also some other Latin name (f.eg. L82 Solanum tuberosum) should be written italics.

All genes name within whole the manuscript should be written italics. Authors very frequently wrongly written about genes function (f.eg. “Apple MdLBD13 gene can inhibit anthocyanin synthesis and nitrogen uptake”), whereas this is cause not by gene per se but by product of expression of this gene – mainly protein. I strongly advise to check whole the manuscript carefully having this into consideration.

The sentence in L41-42: “many LBD protein structures have been found in plants such as rice, apple, and giant salamander” suggest that giant salamander is a plant species. As I know it is an animal – am I right?

I advise you not to used such phrase as in L124-125: “We know that 43 LBD…”, as you know, but the reader not necessary and it should be some reference to this information - your study, literature, database…

Some of the figures as 5 and 6 are not color-blinded friendly.

I advise you to rearrange the figure 10, as A is a control for G, B for H and so on. I will put picture G below A, H below B and so on. The description is also not perfectly clear.

Please correct H2O in L553 into H2O

Author Response

Response to Reviewer 1 Comments

Declaration: All changes in the manuscript are marked in red.

Point 1:

The main remarks concern the statement: “there have been no reports on the potato LBD gene family”. However short literature search give me at least 2 results, which should be mention in this work: the paper by Bernie et al. 2012 (DOI https://doi.org/10.1007/s12230-012-92) published in American Journal of Potato Research and the paper by Bdeir et al. 2016 (DOI https://doi.org/10.1007/s00606-015-12) published in Plant Systematics and Evolution. However they are not so comprehensive look into Solanaceae LBD gene family and possible function of their products, but both of them should be mention. You need to change whole manuscript accordingly.

Response:

Due to previous work mistakes, we did not find the gene “St.LBD” (Unigene Stu.5076 from www.cpgp.ca) published in American Journal of Potato Research by B. J. Zebarth et al 2012. ( DOI https://doi.org/10.1007/s12230-012-92) and the paper by Bdeir et al. 2016(DOI https://doi.org/10.1007/s00606-015-12) ten unnamed LBD genes homologous with AtLBD1, AtLBD4 and AtLBD15 in published in Plant Systematics and Evolution. We apologize for this. We have added these two papers to our paper according to your suggestions. The changes are as follows:

Original manuscript L18-L19:”However, potato LBD protein has not been studied.” changed to L19-L20:“However, the potato systematics and evolution of LBD gene family have not been fully conducted by scholars and researchers.”

Original L387-l388 “but the identification of the potato LBD gene family at the genomic level has not been reported yet” changed L391-L392“However, the identification of the potato LBD gene family at the genomic level has not been fully reported.”

We referenced conclusions in the introduction by Zebarth, B.J, et al. in 2012: L75-L77 ” Zebarth, B.J, et al. find that the gene St.LBD (Unigene Stu.5076 from www.cpgp.ca) is homologous to AtLBD37, AtLBD38 and AtLBD39 in potato may be specifically in response to nitrate rather than N deficiency[27]”

We referenced the conclusions in the Result section that by Bdeir R et al. (DOI https://doi.org/10.1007/s00606-015-12) and discuss L149-L153 ”Interestingly, our results are roughly the same as those of the incomplete potato study by Bdeir R et al[28]. The difference is we have found 7 homologous genes of potato by AtLBD1 which 5 have higher similarities: StLBD4-1, StLBD6-8, StLBD6-2, StLBD11-2, StLBD11-3. 2 similarities are low: StLBD6-11, StLBD1-1. We also find a homologous gene of AtLBD18, it is StLBD1-3, that has not been found by predecessors.”

L315-L317 changed to:” Over time the expression level of StLBD2-6 is similar to StLBD1-5. After 16 days, these two treatments begin to show significant differences and down-regulated expression.”

L341-L342 increased:” The physiological state is shown in Figure 11.”

L348-L349 changed to:” but still retains a certain amount of water has cell viability and has grown tubers (Figure 11B).”

We deleted Original manuscript L419-L421:” Such as StLBD2-6 and StLBD3-5 in the stem, they may have a certain protective effect when the potato is under drought stress to maintain the normal physiological function of the stem.”

We incareased disscution in the Disscution section and modified some sentences L417-L443:” However, the expression level of StLBD3-2 and StLBD11-2 are lower than has been reported transcript data in the leaves which in this study. We found that the expression levels of LBD genes had significant difference in different tissues. The expression of some genes show significant differences under the arid condition. Anthocyanins play a key role in the plant defence [49]. Previous studies, AtLBD37, AtLBD38 and AtLBD39 are negative regulators of anthocyanin synthesis. Overexpression of each of the three genes in the absence of N/NO3- strongly suppresses the key regulators of anthocyanin synthesis [50]. Therefore, we conclude that high expression of these LBD genes is detrimental to increasing plant stress tolerance. DFR gene is an enzyme in the anthocyanin biosynthesis pathway [51]. With the supply of high nitrates, DFR expression is reduced, anthocyanin accumulation is reduced by Zebarth, B.J. et al. This is associated with an increased expression of the N-regulated anthocyanin synthesis inhibitor St.LBD (Unigene Stu. 5076 from www.cpgp.ca). However, ammonium increases has no effect on the expression of St.LBD. Prove that the gene may function specifically in response to nitrate rather than N deficiency. Interestingly, in this study the expression of StLBD1-5 that homologous to AtLBD37, AtLBD38 and AtLBD39 was also down-regulated during drought stress. This may be affecting the absorption of potato to nitrate because of water shortage. We found that the color of potato leaves is deepened under drought (Figure 11) but there is no accumulation of purple matter visible to the naked eye. This shows that the low expression level of StLBD1-5 may increase anthocyanin accumulation in potato leaf to some degree. StLBD1-5 is not the main factor causing anthocyanin accumulation but the high expression of StLBD1-5 is likely to strongly inhibit the accumulation of anthocyanins, thereby reducing the drought resistance of potatoes, which requires further testing. Similarly, StLBD2-6 which has a close relationship with StLBD1-5, they are similar roles in the leaf of potato. The difference is, StLBD2-6 and StLBD3-5 up-regulated expression in stems under drought stress, they may have a certain protective effect when potato under the drought stress to maintain the normal physiological function of the stem. We presume that StLBD2-6 regulates a certain nitrogen metabolism pathway in the stem when potato is subjected to drought stress, in order to maintain the normal physiological metabolism of the stem.”

We deleted original manuscript L421-L424:” It is noteworthy that the potato StLBD2-6 and StLBD1-5 genes are highly homologous to Arabidopsis AtLBD37, AtLBD38 and AtLBD39. Studies have shown that AtLBD37 (AtASL39), AtLBD38 (AtASL40) and AtLBD39 (AtASL41) genes are involved in anthocyanin synthesis and nitrogen metabolism in Arabidopsis”

We deleted original manuscript L427-L430:” Similarly, the transcriptome heat map showed that the potato LBD2-6 gene had the highest expression relative to other potato LBD genes, so it is likely to be involved in the regulation of nitrogen transport in potato plants.”

We deleted original manuscript L469-L471:” Studies have also shown that AtLBD37 (AtASL39), AtLBD38 (AtASL40) and AtLBD39 (AtASL41) genes are involved in anthocyanin synthesis and nitrogen metabolism in Arabidopsis”

original manuscript : “may have similar functions.””changed to new manuscript L482” similar function of nitrate transport.”

Point 2:

Authors used Arabidopsis as a name of Arabidospsis thaliana L. The name written with upper case should be written italics as it is Latin name. If the arabidopsis is used as English name should be written with lower case, however the proper English name is the thale cress. Also some other Latin name (f.eg. L82 Solanum tuberosum) should be written italics.

Response:

According to your suggestion we have changed the name of all Arabidopsis to italics, such as: L95 ” Arabidopsis”. The names of the potatoes are all changed to italics such as L86:” Solanum tuberosum”. Other than this, L43:” Oryza sativa, Malus domestica and Eucalyptus grandis Hill”, L258-L259: ” P. infestans” etc. all have been changed to italics. We have marked them all in red.

point 3:

All genes name within whole the manuscript should be written italics. Authors very frequently wrongly written about genes function (f.eg. “Apple MdLBD13 gene can inhibit anthocyanin synthesis and nitrogen uptake”), whereas this is cause not by gene per se but by product of expression of this gene – mainly protein. I strongly advise to check whole the manuscript carefully having this into consideration.

Response:

According to your suggestion we have changed L69 "MdLBD13" in the " Apple MdLBD13 gene can inhibit anthocyanin synthesis and nitrogen uptake " to italic " MdLBD13 " and carefully checked the whole the manuscript, corrected all such errors. Such as L59: “AtLBD41 (AtASL38)”,L65 “OsAS2”, L71” EgLBD37” etc. We changed all gene names in Table 1 and Table 2 to italics that have been marked in red.

point 4:

The sentence in L41-42: “many LBD protein structures have been found in plants such as rice, apple, and giant salamander” suggest that giant salamander is a plant species. As I know it is an animal am I right?

Response:

Yes, you are right, this is a mistake. We have changed in original manuscript L42-L43: ”giant salamander” to in new manuscript L43: “Eucalyptus grandis Hill

point 5:

I advise you not to used such phrase as in L124-125: “We know that 43 LBD…”, as you know, but the reader not necessary and it should be some reference to this information - your study, literature, database…

Response:

According to your suggestion we have deleted L125-L126 (original manuscript L124-L125): "We know that" from the manuscript changed that to “Analysis of the chromosomal localization results from the potato LBD gene (Figrue 1), 43 LBD genes is distributed into 10 of the 12 chromosomes.”

point 6:

Some of the figures as 5 and 6 are not color-blinded friendly.

Response:

According to your suggestions we changed the color of the expression level heatmap (figure 5 and figure 6) to blue and white (original is red-black-green). Blue represents up-regulation of expression level and white represents down-regulation of expression level. If you have a better suggestion about the color of the heatmap please tell us right away and we will modify it in time.

point 7:

I advise you to rearrange the figure 10, as A is a control for G, B for H and so on. I will put picture G below A, H below B and so on. The description is also not perfectly clear.

Response:

According to your suggestions we put the figure 10 picture G below A, H below B and so on.

We incareased L312-L313: ” On the 16th day, the potato begins to show significant differences under the arid condition (Figure 10K).”

We changed L318-L320 to:” The expression level of StLBD3-1 is very low in leaf, the expression level of StLBD3-1 is the highest on the 12th day under the arid condition and seedling is no lodging at this time (Figure 10J).”

point 8:

Please correct H2O in L553 into H2O.

Response:

We have correced H2O in L560(original L553) into “H2O”.

Other corrections:

Correction 1

We deleted repeated references and ensured that the references are correct.

Correction 2

L239-L244: Change the unit "uM" of to "μM"

Correction 3

We changed the gene names to italics, in both the figure 5 and figure 6.

L202: changed “red-black-green” to “blue and white”, changed” red” to “blue” and so on.

L202: changed “green” to “white” and L254: changed “green” to “white” and so on.

Correction 4

We changed Figure 9 for the following reasons:

First declare that the conclusions we have reached are correct.

In the Figure 9 of original manuscript, we have carried out data conversion that in order to beauty of the picture. However, such show is easy to mislead the reader. The reason is that the expression level of StLBD3-1 is very low, and error caused by the test is higher than the difference. The expression level of StLBD3-1 is very low in any tissues, and there is no such difference in the original picture. So we re-uploaded the figure 9.

Round 2

Reviewer 1 Report

I am horrified to learn that the authors have spent money on translation services for their manuscript. They should demand that their money be returned. The English in the original manuscript was inadequate, and the revision is no better.

Merely in the abstract:

“…the potato systematics and evolution of LBD gene family”

- I’m not sure what “the potato systematics” is trying to say. We know the evolutionary relationships in the Solanum group quite well.

“…the LBD gene existing on potatoes is identified with bioinformatics methods”

- should be “…bioinformatics methods were used to identify the potato LBD gene” – except that it wasn’t just bioinformatics methods – they actually grew the potatoes! And it wasn’t “the” LBD gene, it was 43 of them.

“The number of amino acids coding by the potato LBD gene family ranged…”

- should be “…amino acids encoded by…”

“…under the drought stress”

- no “the”

“enhance the drought resistance against the potato”

- I think they mean “…enhance the potatoes’ resistance to drought”.

I can’t go through the entire manuscript like this. The authors need to find a native (or at least fluent) English speaker to help them out. And they need to never use the translation service they paid again!

Author Response

Response to Reviewer 1 Comments

Declaration: All changes in the manuscript are marked in red.

Comment 1:

I am horrified to learn that the authors have spent money on translation services for their manuscript. They should demand that their money be returned. The English in the original manuscript was inadequate, and the revision is no better.Merely in the abstract:

“…the potato systematics and evolution of LBD gene family”

- I’m not sure what “the potato systematics” is trying to say. We know the evolutionary relationships in the Solanum group quite well.

Response:

We are so glad to receive your letter! We edited the English again and hope to meet your requirements.

According to your suggestion, we have changed“…the potato systematics and evolution of LBD gene family” to L19-L20: “However, a potato phylogenetic analysis of the LBD family has not been fully studied by scholars and researchers.”

Comment 2:

“…the LBD gene existing on potatoes is identified with bioinformatics methods”

- should be “…bioinformatics methods were used to identify the potato LBD gene” – except that it wasn’t just bioinformatics methods – they actually grew the potatoes! And it wasn’t “the” LBD gene, it was 43 of them.

Response:

According to your suggestion, we have changed “…the LBD gene existing on potatoes is identified with bioinformatics methods” to L20-L21: “In this research, bioinformatics methods and the growth of potatoes were used to identify 43 StLBD proteins.”

Comment 3:

“The number of amino acids coding by the potato LBD gene family ranged…”

- should be “…amino acids encoded by…”

Response:

According to your suggestion, we have changed “The number of amino acids coding by the potato LBD gene family ranged…” to L22-L23: “The number of amino acids encoded by the potato LBD family ranged from 94 to 327.”

Comment 4:

“…under the drought stress”- no “the”

Response:

According to your suggestion, we have changed “…under the drought stress” to L27: “down-regulated under drought stress,” and we checked all manuscripts, corrected such errors.

Comment 5:

“…enhance the drought resistance against the potato”

- I think they mean “…enhance the potatoes’ resistance to drought”.

Response:

According to your suggestion, we have changed “…enhance the drought resistance against the potato” to L28: “…enhance the potatoes’ resistance to drought.”

Comment 6:

I can’t go through the entire manuscript like this. The authors need to find a native (or at least fluent) English speaker to help them out.

Response:

We have edited English again and will continue to modify. Thank you for forgiving!

Other corrections:

Correction 1

Figure 5: We deleted the redundant “leaf”.

Correction 2

Figure 9: Iincreased “A, B and C”

Reviewer 2 Report

Reviewer would like to thank to the author for their replay and substantial improvement in manuscript file. 

Some minor remarks:

the literature under no 27 and 49 is doubled;

my remarks on gene function is still repeated. From physiological and biochemical point of view proteins coded by particular genes are responsible for their function as genes per se do not play role in metabolic reaction, do not catalyse reaction and mostly have not enzymatic activity. There are proteins coded by this genes.

f.eg. "LBD10 plays a 63 crucial role in the development of Arabidopsis pollen[20]." - gene or protein?

"AtLBD6 (AtAS2) inhibits cell 61 proliferation and near-distal axis symmetry in the paraxial region of the leaf forming flattened leaf 62 and regulating flower development through synergy with AS1 and JAG[18, 19]." - gene or product of expression of this gene?

I leave this remarks under consideration by the authors.

Author Response

Response to Reviewer 2 Comments

Declaration: All changes in the manuscript are marked in red.

Comment 1:

The literature under no 27 and 49 is doubled;

Response:

We have deleted the reference “[49]”, thank you very much!

Comment 2:

My remarks on gene function is still repeated. From physiological and biochemical point of view proteins coded by particular genes are responsible for their function as genes per se do not play role in metabolic reaction, do not catalyse reaction and mostly have not enzymatic activity. There are proteins coded by this genes.

Response:

We are so glad to receive your letter!

You are right. Genes perse do not play role in metabolic reaction, while “LBD” are refer in particular to “transcription factors of LOB (lateral organ boundaries) specific domains”, which are transcription factors and are proteins. “LOB proves to be a nuclear protein and involved in transcriptional regulation” (https://doi.org/10.1007/s11738-009-0375-3). In this study, we have studied from genomic levels, transcription levels and protein levels to study LBD family members. Genomic levels like: Figure 1 shows the location of the LBD genes on the chromosome in potato. Figure 3 (b) shows gene structure and Figure 7 shows Cis-acting element. Transcription levels like: Figure 5 shows the expression profiles of StLBD genes in different tissues of potato. Figure 6 shows the expression profile of potato LBD genes under ten different biotic or abiotic stress. Figures 8 and 9 are also analyzed from the transcription levels. Protein levels like: Table 1 shows protein features (Protein length etc.). Figure 2 shows phylogenetic tree of LBD transcription factors. Figure 3 (a) and (c) shows conserved domain. Figure 4 shows proteins sequence alignment. However, all the above is for a more in-depth study of the LBD transcription factor (protein). We can alter plants by regulating gene expression to regulate LBD transcription factors. Therefore, this research is important.

Comment 3:

"LBD10 plays a crucial role in the development of Arabidopsis pollen [20]." - Gene or protein?

Response:

According to your suggestion we have changed “LBD10 plays a 63 crucial role in the development of Arabidopsis pollen [20]” to L65-L66: “The LBD10 protein plays a crucial role in the development of Arabidopsis pollen [20].”

Comment4:

"AtLBD6 (AtAS2) inhibits cell proliferation and near-distal axis symmetry in the paraxial region of the leaf forming flattened leaf and regulating flower development through synergy with AS1 and JAG[18, 19]." - gene or product of expression of this gene?

Response:

According to your suggestion we have changed “tLBD6 (AtAS2) inhibits cell proliferation and near-distal axis symmetry in the paraxial region of the leaf forming flattened leaf and regulating flower development through synergy with AS1 and JAG[18, 19]” to L63-L65:“The expression product of AtLBD6 (AtAS2) inhibits cell proliferation and near-distal axis symmetry in the paraxial region of the leaf, forming a flattened leaf and regulating flower development through synergy with AS1 and JAG [18, 19]”

Other corrections:

Correction 1

Figure 5: We deleted the redundant “leaf”.

Correction 2

Figure 9: Iincreased “A, B and C”.

Round 3

Reviewer 1 Report

The English is fine now. The authors are too be commended on what I'm sure was an enormous amount of work to get the manuscript to this stage.